# Real-Time Bidding with Side Information

**Arthur Flajolet**
MIT, ORC
flajolet@mit.edu

**Patrick Jaillet**
MIT, EECS, LIDS, ORC
jaillet@mit.edu

## Abstract

We consider the problem of repeated bidding in online advertising auctions when some side information (e.g. browser cookies) is available ahead of submitting a bid in the form of a $d$-dimensional vector. The goal for the advertiser is to maximize the total utility (e.g. the total number of clicks) derived from displaying ads given that a limited budget $B$ is allocated for a given time horizon $T$. Optimizing the bids is modeled as a contextual Multi-Armed Bandit (MAB) problem with a knapsack constraint and a continuum of arms. We develop UCB-type algorithms that combine two streams of literature: the confidence-set approach to linear contextual MABs and the probabilistic bisection search method for stochastic root-finding. Under mild assumptions on the underlying unknown distribution, we establish distribution-independent regret bounds of order $\tilde{O}(d \cdot \sqrt{T})$ when either $B = \infty$ or when $B$ scales linearly with $T$.

## 1 Introduction

On the internet, advertisers and publishers now interact through real-time marketplaces called ad exchanges. Through them, any publisher can sell the opportunity to display an ad when somebody is visiting a webpage he or she owns. Conversely, any advertiser interested in such an opportunity can pay to have his or her ad displayed. In order to match publishers with advertisers and to determine prices, ad exchanges commonly use a variant of second-price auctions which typically runs as follows. Each participant is initially provided with some information about the person that will be targeted by the ad (e.g. browser cookies, IP address, and operating system) along with some information about the webpage (e.g. theme) and the ad slot (e.g. width and visibility). Based on this limited knowledge, advertisers must submit a bid in a timely fashion if they deem the opportunity worthwhile. Subsequently, the highest bidder gets his or her ad displayed and is charged the second-highest bid. Moreover, the winner can usually track the customer's interaction with the ad (e.g. clicks). Because the auction is sealed, very limited feedback is provided to the advertiser if the auction is lost. In particular, the advertiser does not receive any customer feedback in this scenario. In addition, the demand for ad slots, the supply of ad slots, and the websurfers' profiles cannot be predicted ahead of time and are thus commonly modeled as random variables, see [19]. These two features contribute to making the problem of bid optimization in ad auctions particularly challenging for advertisers.

### 1.1 Problem statement and contributions

We consider an advertiser interested in purchasing ad impressions through an ad exchange. As standard practice in the online advertising industry, we suppose that the advertiser has allocated a limited budget $B$ for a limited period of time, which corresponds to the next $T$ ad auctions. Rounds, indexed by $t \in \mathbb{N}$, correspond to ad auctions in which the advertiser participates. At the beginning of round $t \in \mathbb{N}$, some contextual information about the ad slot and the person that will be targeted is revealed to the advertiser in the form of a multidimensional vector $x_t \in \mathcal{X}$, where $\mathcal{X}$ is a subset of $\mathbb{R}^d$. Without loss of generality, the coordinates of $x_t$ are assumed to be normalized in such a way that

$\|x\|_{\infty} \leq 1$ for all $x \in \mathcal{X}$. Given $x_t$, the advertiser must submit a bid $b_t$ in a timely fashion. If $b_t$ is larger than the highest bid submitted by the competitors, denoted by $p_t$ and also referred to as the market price, the advertiser wins the auction, is charged $p_t$, and gets his or her ad displayed, from which he or she derives a utility $v_t$. Monetary amounts and utility values are assumed to be normalized in such a way that $b_t, p_t, v_t \in [0, 1]$. In this modeling, one of the competitors is the publisher himself who submits a reserve price so that $p_t > 0$. No one wins the auction if no bid is larger than the reserve price. For the purpose of modeling, we suppose that ties are broken in favor of the advertiser but this choice is arbitrary and by no means a limitation of the approach. Hence, the advertiser collects a reward $r_t = v_t \cdot \mathbb{1}_{b_t \geq p_t}$ and is charged $c_t = p_t \cdot \mathbb{1}_{b_t \geq p_t}$ at the end of round $t$. Since the monetary value of getting an ad displayed is typically difficult to assess, $v_t$ and $c_t$ may be expressed in different units and thus cannot be compared directly in general, which makes the problem two-dimensional. This is the case, for example, when the goal of the advertiser is to maximize the number of clicks, in which case $v_t = 1$ if the ad was clicked on and $v_t = 0$ otherwise. We consider a stochastic setting where the environment and the competitors are not fully adversarial. Specifically, we assume that, at any round $t \in \mathbb{N}$, the vector $(x_t, v_t, p_t)$ is jointly drawn from a fixed probability distribution $\nu$ independently from the past. While this assumption may seem unnatural at first as the other bidders also act as learning agents, it is motivated by the following observation. In our setting, we consider that there are many bidders, each participating in a small subset of a large number of auctions, that value ad opportunities very differently depending on the intended audience, the nature and topic of the ads, and other technical constraints. Since bidders have no idea who they will be competing against for a particular ad (because the auctions are sealed), they are naturally led to be oblivious to the competition and to bid with the only objective of maximizing their own objective functions. Given the variety of objective functions and the large number of bidders and ad auctions, we argue that, by the law of large numbers, the process $(x_t, p_t, v_t)_{t=1,\dots,T}$ that we experience as a bidder is i.i.d., at least for a short period of time. Moreover, while the assumption that the distribution of $(x_t, v_t, p_t)$ is stationary may only be valid for a short period of time, advertisers tend to participate in a large number of ad auctions per second so that $T$ and $B$ are typically large values, which motivates an asymptotic study. We generically denote by $(X, V, P)$ a vector of random variables distributed according to $\nu$. We make a structural assumption about $\nu$, which we use throughout the paper.

**Assumption 1.** *The random variables $V$ and $P$ are conditionally independent given $X$. Moreover, there exists $\theta_* \in \mathbb{R}^d$ such that $\mathbb{E}[V \mid X] = X^{\intercal}\theta_*$ and $\|\theta_*\|_{\infty} \leq 1$.*

Note, in particular, that Assumption 1 is satisfied if $V$ and $P$ are deterministic functions of $X$. The first part of Assumption 1 is very natural since: (i) $X$ captures all and only the information about the ad shared to all bidders before submitting a bid and (ii) websurfers are oblivious to the ad auctions that take place behind the scenes to determine which ad they will be presented with. The second part of Assumption 1 is standard in the literature on linear contextual MABs, see [1] and [16], and is arguably the simplest model capturing a dependence between $x_t$ and $v_t$. When the advertiser's objective is to maximize the number of clicks, this assumption translates into a linear Click-Through Rate (CTR) model.

We denote by $(\mathcal{F}_t)_{t \in \mathbb{N}}$ (resp. $(\tilde{\mathcal{F}}_t)_{t \in \mathbb{N}}$) the natural filtration generated by $((x_t, v_t, p_t))_{t \in \mathbb{N}}$ (resp. $((x_{t+1}, v_t, p_t))_{t \in \mathbb{N}}$). Since the advertiser can keep bidding only so long as he or she does not run out of money or time, he or she can no longer participate in ad auctions at round $\tau^*$, mathematically defined by:

$$\tau^* = \min(T+1, \min\{t \in \mathbb{N} \mid \sum_{\tau=1}^{t} c_\tau > B\}).$$

Note that $\tau^*$ is a stopping time with respect to $(\mathcal{F}_t)_{t \in \mathbb{N}}$. The difficulty for the advertiser when it comes to determining how much to bid at each round lies in the fact that the underlying distribution $\nu$ is initially unknown. This task is further complicated by the fact that the feedback provided to the advertiser upon bidding $b_t$ is partially censored: $p_t$ and $v_t$ are only revealed if the advertiser wins the auction, i.e. if $b_t \geq p_t$. In particular when $b_t < p_t$, the advertiser can never evaluate how much reward would have been obtained and what price would have been charged if he or she had submitted a higher bid. The goal for the advertiser is to design a non-anticipating algorithm that, at any round $t$, selects $b_t$ based on the information acquired in the past so as to keep the pseudo-regret defined as:

$$R_{B,T} = \mathrm{ER}_{\mathrm{OPT}}(B, T) - \mathbb{E}[\sum_{t=1}^{\tau^*-1} r_t]$$

as small as possible, where $\text{ER}_{\text{OPT}}(B,T)$ is the maximum expected sum of rewards that can be obtained by a non-anticipating oracle algorithm that has knowledge of the underlying distribution. Here, an algorithm is said to be non-anticipating if the bid selection process does not depend on the future observations. We develop algorithms with bounds on the pseudo-regret that do not depend on the underlying distribution $\nu$, which are referred to as distribution-independent regret bounds. This entails studying the asymptotic behavior of $R_{B,T}$ when $B$ and $T$ go to infinity. For mathematical convenience, we consider that the advertiser keeps bidding even if he or she has run out of time or money so that all quantities are well defined for any $t \in \mathbb{N}$. Of course, the rewards obtained for $t \geq \tau^*$ are not taken into account in the advertiser's total reward when establishing regret bounds.

**Contributions** We develop UCB-type algorithms that combine the ellipsoidal confidence set approach to linear contextual MAB problems with a special-purpose stochastic binary search procedure. When the budget is unlimited or when it scales linearly with time, we show that, under additional technical assumptions on the underlying distribution $\nu$, our algorithms incur a regret $R_{B,T} = \tilde{O}(d \cdot \sqrt{T})$, where the $\tilde{O}$ notation hides logarithmic factors in $d$ and $T$. A key insight is that overbidding is not only essential to incentivize exploration in order to estimate $\theta_*$, but also crucial to find the optimal bidding strategy given $\theta_*$ because bidding higher always provide more feedback in real-time bidding.

## 1.2 Literature review

To handle the exploration-exploitation trade-off inherent to MAB problems, an approach that has proved to be particularly successful is the *optimism in the face of uncertainty* paradigm. The idea is to consider all plausible scenarios consistent with the information collected so far and to select the decision that yields the largest reward among all identified scenarios. Auer et al. [7] use this idea to solve the standard MAB problem where decisions are represented by $K \in \mathbb{N}$ arms and pulling arm $k \in \{1, \cdots, K\}$ at round $t \in \{1, \cdots, T\}$ yields a random reward drawn from an unknown distribution specific to this arm independently from the past. Specifically, Auer et al. [7] develop the Upper Confidence Bound algorithm (UCB1), which consists in selecting the arm with the current largest upper confidence bound on its mean reward, and establish near-optimal regret bounds. This approach has since been successfully extended to a number of more general settings. Of most notable interest to us are: (i) linear contextual MAB problems, where, for each arm $k$ and at each round $t$, some context $x_t^k$ is provided to the decision maker ahead of pulling any arm and the expected reward of arm $k$ is $\theta_*^\top x_t^k$ for some unknown $\theta_* \in \mathbb{R}^d$, and (ii) the Bandits with Knapsacks (BwK) framework, an extension to the standard MAB problem allowing to model resource consumption.

UCB-type algorithms for linear contextual MAB problems were first developed in [6] and later extended and improved upon in [1] and [16]. In this line of work, the key idea is to build, at any round $t$, an ellipsoidal confidence set $\mathcal{C}_t$ on the unknown parameter $\theta_*$ and to pull the arm $k$ that maximizes $\max_{\theta \in \mathcal{C}_t} \theta^\top x_t^k$. Using this idea, Chu et al. [16] derive $\tilde{O}(\sqrt{d \cdot T})$ upper bounds on regret that hold with high probability, where the $\tilde{O}$ notations hides logarithmic factors in $d$ and $T$. While this result is not directly applicable in our setting, partly because of the knapsack constraint, we rely on this technique to estimate $\theta_*$.

The real-time bidding problem considered in this work can be formulated as a BwK problem with contextual information and a continuum of arms. This framework, first introduced in its full generality in [10] and later extended to incorporate contextual information in [11], [3], and [2], captures resource consumption by assuming that pulling any arm incurs the consumption of possibly many different limited resource types by random amounts. BwK problems are notoriously harder to solve than standard MAB problems. For example, sublinear regret cannot be achieved in general for BwK problems when an opponent is adversarially picking the rewards and the amounts of resource consumption at each round, see [10], while this is possible for standard MAB problems, see [8]. The problem becomes even more complex when some contextual information is available at the beginning of each round as approaches developed for standard contextual MAB problems and for BwK problems fail when applied to contextual BwK problems, see the discussion in [11], which calls for the development of new techniques. Agrawal and Devanur [2] consider a particular case where the expected rewards and the expected amounts of resource consumption are linear in the context and derive, in particular, $\tilde{O}(\sqrt{d \cdot T})$ bounds on regret when the initial endowments of resources scale linearly with the time horizon $T$. These results do not carry over

to our setting because the expected costs, and in fact also the expected rewards, are not linear in the context. To the best of our knowledge, the only prior works that deal simultaneously with knapsack constraints and a non-linear dependence of the rewards and the amounts of resource consumption on the contextual information are Agrawal et al. [3] and Badanidiyuru et al. [11]. When there is a finite number of arms $K$, they derive regret bounds that scale as $\tilde{O}(\sqrt{K \cdot T \cdot \ln(\Pi)})$, where $\Pi$ is the size of the set of benchmark policies. To some extent, at least when $\theta_*$ is known, it is possible to apply these results but this requires to discretize the set of valid bids $[0, 1]$ and the regret bounds thus derived scale as $\sim T^{2/3}$, see the analysis in [10], which is suboptimal.

On the modeling side, the most closely related prior works studying repeated ad auctions under the lens of online learning are [25], [23], [17], [12], and [5]. Weed et al. [25] develop algorithms to solve the problem considered in this work when no contextual information is available and when there is no budget constraint, in which case the rewards are defined as $r_t = (v_t - p_t) \cdot \mathbb{1}_{b_t \geq p_t}$, but in a more general adversarial setting where few assumptions are made concerning the sequence $((v_t, p_t))_{t \in \mathbb{N}}$. They obtain $\tilde{O}(\sqrt{T})$ regret bounds with an improved rate $O(\ln(T))$ in some favorable settings of interest. Inspired by [4], Tran-Thanh et al. [23] study a particular case of the problem considered in this work when no contextual information is available and when the goal is to maximize the number of impressions. They use a dynamic programming approach and claim to derive $\tilde{O}(\sqrt{T})$ regret bounds. Balseiro and Gur [12] identify near-optimal bidding strategies in a game-theoretic setting assuming that each bidder has a black-box function that maps the contextual information available before bidding to the expected utility derived from displaying an ad (which amounts to assuming that $\theta_*$ is known a priori in our setting). They show that bidding an amount equal to the expected utility derived from displaying an ad normalized by a bid multiplier, to be estimated, is a near-optimal strategy. We extend this observation to the contextual settings. Compared to their work, the difficulty in our setting lies in estimating simultaneously the bid multiplier and $\theta_*$. Finally, the authors of [5] and [17] take the point of view of the publisher whose goal is to price ad impressions, as opposed to purchasing them, in order to maximize revenues with no knapsack constraint. Cohen et al. [17] derive $O(\ln(d^2 \cdot \ln(T/d)))$ bounds on regret with high probability with a multidimensional binary search.

On the technical side, our work builds upon and contributes to the stream of literature on probabilistic bisection search algorithms. This class of algorithms was originally developed for solving stochastic root finding problems, see [22] for an overview, but has also recently appeared in the MAB literature, see [20]. Our approach is largely inspired by the work of Lei et al. [20] who develop a stochastic binary search algorithm to solve a dynamic pricing problem with limited supply but no contextual information, which can be modeled as a BwK problem with a continuum of arms. Dynamic pricing problems with limited supply are often modeled as BwK problems in the literature, see [24], [9], and [20], but, to the best of our knowledge, the availability of contextual information about potential customers is never captured. Inspired by the technical developments introduced in these works, our approach is to characterize a near-optimal strategy in closed form and to refine our estimates of the (usually few) initially unknown parameters involved in the characterization as we make decisions online, implementing this strategy using the latest estimates for the parameters. However, the technical challenge in these works differs from ours in one key aspect: the feedback provided to the decision maker is completely censored in dynamic pricing problems, since the customers' valuations are never revealed, while it is only partially censored in real-time bidding, since the market price is revealed if the auction is won. Making the most of this additional feature enables us to develop a stochastic binary search procedure that can be compounded with the ellipsoidal confidence set approach to linear contextual bandits in order to incorporate contextual information.

**Organization**  The remainder of the paper is organized as follows. In order to increase the level of difficulty progressively, we start by studying the situation of an advertiser with unlimited budget, i.e. $B = \infty$, in Section 2. Given that second-price auctions induce truthful bidding when the bidder has no budget constraint, this setting is easier since the optimal bidding strategy is to bid $b_t = x_t^\top \theta_*$ at any round $t \in \mathbb{N}$. This drives us to focus on the problem of estimating $\theta_*$, which we do by means of ellipsoidal confidence sets. Next, in Section 3, we study the setting where $B$ is finite and scales linearly with the time horizon $T$. We show that a near-optimal strategy is to bid $b_t = x_t^\top \theta_* / \lambda_*$ at any round $t \in \mathbb{N}$, where $\lambda_* \geq 0$ is a scalar factor whose purpose is to spread the budget as evenly as possible, i.e. $\mathbb{E}[P \cdot \mathbb{1}_{X^\top \theta_* \geq \lambda_* \cdot P}] = B/T$. Given this characterization, we first assume that $\theta_*$

is known a priori to focus instead on the problem of computing an approximate solution $\lambda \geq 0$ to $\mathbb{E}[P \cdot \mathbb{1}_{X^\intercal \theta_* \geq \lambda \cdot P}] = B/T$ in Section 3.1. We develop a stochastic binary search algorithm for this purpose which is shown to incur $\tilde{O}(\sqrt{T})$ regret under mild assumptions on the underlying distribution $\nu$. In Section 3.2, we bring the stochastic binary search algorithm together with the estimation method based on ellipsoidal confidence sets to tackle the general problem and derive $\tilde{O}(d \cdot \sqrt{T})$ regret bounds. All the proofs are deferred to the Appendix.

**Notations** For a vector $x \in \mathbb{R}^d$, $\|x\|_\infty$ refers to the $L_\infty$-norm of $x$. For a positive definite matrix $M \in \mathbb{R}^{d \times d}$ and a vector $x \in \mathbb{R}^d$, we define the norm $\|x\|_M$ as $\|x\|_M = \sqrt{x^\intercal M x}$. For $x, y \in \mathbb{R}^d$, it is well known that the following Cauchy-Schwarz inequality holds: $|x^\intercal y| \leq \|x\|_M \cdot \|y\|_{M^{-1}}$. We denote by $I_d$ the identity matrix in dimension $d$. We use the standard asymptotic notation $O(\cdot)$ when $T$, $B$, and $d$ go to infinity. We also use the notation $\tilde{O}(\cdot)$ that hides logarithmic factors in $d$, $T$, and $B$. For $x \in \mathbb{R}$, $(x)_+$ refers to the positive part of $x$. For a finite set $S$ (resp. a compact interval $I \subset \mathbb{R}$), $|S|$ (resp. $|I|$) denotes the cardinality of $S$ (resp. the length of $I$). For a set $S$, $\mathcal{P}(S)$ denotes the set of all subsets of $S$. Finally, for a real-valued function $f(\cdot)$, $\operatorname{supp} f(\cdot)$ denotes the support of $f(\cdot)$.

## 2 Unlimited budget

In this section, we suppose that the budget is unlimited, i.e. $B = \infty$, which implies that the rewards have to be redefined in order to directly incorporate the costs. For this purpose, we assume in this section that $v_t$ is expressed in monetary value and we redefine the rewards as $r_t = (v_t - p_t) \cdot \mathbb{1}_{b_t \geq p_t}$. Since the budget constraint is irrelevant when $B = \infty$, we use the notations $R_T$ and $\mathrm{ER}_{\mathrm{OPT}}(T)$ in place of $R_{B,T}$ and $\mathrm{ER}_{\mathrm{OPT}}(B, T)$. As standard in the literature on MAB problems, we start by analyzing the optimal oracle strategy that has knowledge of the underlying distribution. This will not only guide the design of algorithms when $\nu$ is unknown but this will also facilitate the regret analysis. The algorithm developed in this section as well as the regret analysis are extensions of the work of Weed et al. [25] to the contextual setting.

**Benchmark analysis** It is well known that second-price auctions induce truthful bidding in the sense that any participant whose only objective is to maximize the immediate payoff should always bid what he or she thinks the good being auctioned is worth. The following result should thus come at no surprise in the context of real-time bidding given Assumption 1 and the fact that each participant is provided with the contextual information $x_t$ before the $t$-th auction takes place.

**Lemma 1.** *The optimal non-anticipating strategy is to bid $b_t = x_t^\intercal \theta_*$ at any time period $t \in \mathbb{N}$ and we have $\mathrm{ER}_{\mathrm{OPT}}(T) = \sum_{t=1}^T \mathbb{E}[(x_t^\intercal \theta_* - p_t)_+]$.*

Lemma 1 shows that the problem faced by the advertiser essentially boils down to estimating $\theta_*$. Since the bidder only gets to observe $v_t$ if the auction is won, this gives advertisers a clear incentive to overbid early on so that they can progressively refine their estimates downward as they collect more data points.

**Specification of the algorithm** Following the approach developed in [6] for linear contextual MAB problems, we define, at any round $t$, the regularized least square estimate of $\theta_*$ given all the feedback acquired in the past $\hat{\theta}_t = M_t^{-1} \sum_{\tau=1}^{t-1} \mathbb{1}_{b_\tau \geq p_\tau} \cdot v_\tau \cdot x_\tau$, where $M_t = I_d + \sum_{\tau=1}^{t-1} \mathbb{1}_{b_\tau \geq p_\tau} \cdot x_\tau x_\tau^\intercal$, as well as the corresponding ellipsoidal confidence set:

$$\mathcal{C}_t = \{\theta \in \mathbb{R}^d \mid \left\|\theta - \hat{\theta}_t\right\|_{M_t} \leq \delta_T\},$$

with $\delta_T = 2\sqrt{d \cdot \ln((1 + d \cdot T) \cdot T)}$. For the reasons mentioned above, we take the *optimism in the face of uncertainty approach* and bid:

$$b_t = \max(0, \min(1, \max_{\theta \in \mathcal{C}_t} \theta^\intercal x_t)) = \max(0, \min(1, \hat{\theta}_t^\intercal x_t + \delta_T \cdot \sqrt{x_t^\intercal M_t^{-1} x_t})) \tag{1}$$

at any round $t$. Since $\mathcal{C}_t$ was designed with the objective of guaranteeing that $\theta_* \in \mathcal{C}_t$ with high probability at any round $t$, irrespective of the number of auctions won in the past, $b_t$ is larger than the optimal bid $x_t^\intercal \theta_*$ in general, i.e. we tend to overbid.

**Regret analysis** Concentration inequalities are intrinsic to any kind of learning and are thus key to derive regret bounds in online learning. We start with the following lemma, which is a consequence of the results derived in [1] for linear contextual MABs, that shows that $\theta_*$ lies in all the ellipsoidal confidence sets with high probability. Assumption 1 is key to establish this result.

**Lemma 2.** *We have* $\mathbb{P}[\theta^* \notin \cap_{t=1}^T \mathcal{C}_t] \leq \frac{1}{T}$.

Equipped with Lemma 2 along with some standard results for linear contextual bandits, we are now ready to extend the analysis of Weed et al. [25] to the contextual setting.

**Theorem 1.** *Bidding according to* (1) *incurs a regret* $R_T = \tilde{O}(d \cdot \sqrt{T})$.

**Alternative algorithm with lazy updates** As first pointed out by Abbasi-Yadkori et al. [1] in the context of linear bandits, updating the confidence set $\mathcal{C}_t$ at every round is not only inefficient but also unnecessary from a performance standpoint. Instead, we can perform batch updates, only updating $\mathcal{C}_t$ using all the feedback collected in the past at rounds $t$ for which $\det(M_t)$ has increased by a factor at least $(1 + A)$ compared to the last time there was an update, for some constant $A > 0$ of our choosing. This leads to an interesting trade-off between computational efficiency and deterioration of the regret bound captured in our next result. For mathematical convenience, we keep the same notations as when we were updating the confidence sets at every round. The only difference lies in the fact that the bid submitted at time $t$ is now defined as:

$$b_t = \max(0, \min(1, \max_{\theta \in \mathcal{C}_{\tau_t}} \theta^\intercal x_t)), \tag{2}$$

where $\tau_t$ is the last round before round $t$ where the last batch update happened.

**Theorem 2.** *Bidding according to* (2) *at any round* $t$ *incurs a regret* $R_T = \tilde{O}(d \cdot \sqrt{A \cdot T})$.

The fact that we can afford lazy updates will turn out to be important to tackle the general case in Section 3.2 since we will only be able to update the confidence sets at most $O(\ln(T))$ times overall.

## 3 Limited budget

In this section, we consider the setting where $B$ is finite and scales linearly with the time horizon $T$. We will need the following assumptions for the remainder of the paper.

**Assumption 2.** *(a)* $B/T = \beta$ *is a constant independent of any other relevant quantities.*

*(b) There exists* $r > 0$, *known to the advertiser, such that* $p_t \geq r$ *for all* $t \in \mathbb{N}$.

*(c) We have* $\mathbb{E}[1/X^\intercal \theta_*] < \infty$.

*(d) The random variable* $P$ *has a continuous conditional probability density function given the occurrence of the value* $x$ *of* $X$, *denoted by* $f_x(\cdot)$, *that is upper bounded by* $\bar{L} < \infty$.

Conditions (a) and (b) are very natural in real-time bidding where the budget scales linearly with time and where $r$ corresponds to the minimum reserve price across ad auctions. Observe that Condition (c) is satisfied, for example, when the probability of a click given any context is at least no smaller than a (possibly unknown) positive threshold. Condition (d) is motivated by technical considerations that will appear clear in the analysis. Note that $\bar{L}$ is not assumed to be known to the advertiser.

In order to increase the level of difficulty progressively and to prepare for the integration of the ellipsoidal confidence sets, we first look at an artificial setting in Section 3.1 where we assume that there exists a known set $\mathcal{C} \subset \mathbb{R}^d$ such that $\mathbb{E}[V|X] = \min(1, \max_{\theta \in \mathcal{C}} X^\intercal \theta)$ (as opposed to $\mathbb{E}[V|X] = X^\intercal \theta_*$) and such that $\theta_* \in \mathcal{C}$. This is to sidestep the estimation problem in a first step in order to focus on determining an optimal bidding strategy given $\theta_*$. Next, in Section 3.2, we bring together the methods developed in Section 2 and Section 3.1 to tackle the general setting.

### 3.1 Preliminary work

In this section, we make the following modeling assumption in lieu of $\mathbb{E}[V|X] = X^\intercal \theta_*$.

**Assumption 3.** *There exists* $\mathcal{C} \subset \mathbb{R}^d$ *such that* $\mathbb{E}[V|X] = \min(1, \max_{\theta \in \mathcal{C}} X^\intercal \theta)$ *and* $\theta_* \in \mathcal{C}$.

Furthermore, we assume that $\mathcal{C}$ is known to the advertiser initially. Of course, we recover the original setting introduced in Section 1 when $\mathcal{C} = \{\theta_*\}$ (since $V \in [0, 1]$ implies $\mathbb{E}[V|X] \in [0, 1]$) and $\theta_*$ is known but the level of generality considered here will prove useful to tackle the general case in Section 3.2 when we define $\mathcal{C}$ as an ellipsoidal confidence set on $\theta_*$. As in Section 2, we start by identifying a near-optimal oracle bidding strategy that has knowledge of the underlying distribution. This will not only guide the design of algorithms when $\nu$ is unknown but this will also facilitate the regret analysis. We use the shorthand $g(X) = \min(1, \max_{\theta \in \mathcal{C}} X^\mathsf{T}\theta)$ throughout this section.

**Benchmark analysis**    To bound the performance of any non-anticipating strategy, we will be interested in the mappings $\phi : \lambda, \mathcal{C} \to \mathbb{E}[P \cdot \mathbb{1}_{g(X) \geq \lambda \cdot P}]$ and $R : \lambda, \mathcal{C} \to \mathbb{E}[g(X) \cdot \mathbb{1}_{g(X) \geq \lambda \cdot P}]$ for $(\lambda, \mathcal{C}) \in [0, 2/r] \times \mathcal{P}(\mathbb{R}^d)$. Note that $\phi(\cdot, \mathcal{C})$ is non-increasing and that, without loss of generality, we can restrict $\lambda$ to be no larger than $2/r$ because $\phi(\lambda, \mathcal{C}) = \phi(2/r, \mathcal{C}) = 0$ for $\lambda \geq 2/r$ since $P \geq r$. Exploiting the structure of the MAB problem at hand, we can bound the sum of rewards obtained by any non-anticipating strategy by the value of a knapsack problem where the weights and the values of the items are drawn in an i.i.d. fashion from a fixed distribution. Since characterizing the expected optimal value of a knapsack problem is a well-studied problem, see [21], we can derive a simple upper bound on $\mathrm{ER}_{\mathrm{OPT}}(B, T)$ through this reduction, as we next show.

**Lemma 3.** *We have $\mathrm{ER}_{\mathrm{OPT}}(B, T) \leq T \cdot R(\lambda_*, \mathcal{C}) + \sqrt{T}/r + 1$, where $\lambda_* \geq 0$ satisfies $\phi(\lambda_*, \mathcal{C}) = \beta$ or $\lambda_* = 0$ if no such solution exists (i.e. if $\mathbb{E}[P] < \beta$) in which case $\phi(\lambda_*, \mathcal{C}) \leq \beta$.*

Lemma 3 suggests that, given $\mathcal{C}$, a good strategy is to bid $b_t = \min(1, \min(1, \max_{\theta \in \mathcal{C}} x_t^\mathsf{T}\theta)/\lambda_*)$, at any round $t$. The following result shows that we can actually afford to settle for an approximate solution $\lambda \geq 0$ to $\phi(\lambda, \mathcal{C}) = \beta$.

**Lemma 4.** *For any $\lambda_1, \lambda_2 \geq 0$, we have: $|R(\lambda_1, \mathcal{C}) - R(\lambda_2, \mathcal{C})| \leq 1/r \cdot |\phi(\lambda_1, \mathcal{C}) - \phi(\lambda_2, \mathcal{C})|$.*

Lemma 3 combined with Lemma 4 suggests that the problem of computing a near-optimal bidding strategy essentially reduces to a stochastic root-finding problem for the function $|\phi(\cdot, \mathcal{C}) - \beta|$. As it turns out, the fact that the feedback is only partially censored makes a stochastic bisection search possible with minimal assumptions on $\phi(\cdot, \mathcal{C})$. Specifically, we only need that $\phi(\cdot, \mathcal{C})$ be Lipschitz, while the technique developed in [20] for a dynamic pricing problem requires $\phi(\cdot, \mathcal{C})$ to be bi-Lipschitz. This is a significant improvement because this last condition is not necessarily satisfied uniformly for all confidence sets $\mathcal{C}$, which will be important when we use a varying ellipsoidal confidence set instead of $\mathcal{C} = \{\theta_*\}$ in Section 3.2. Note, however, that Assumption 2 guarantees that $\phi(\cdot, \mathcal{C})$ is always Lipschitz, as we next show.

**Lemma 5.** $\phi(\cdot, \mathcal{C})$ *is $\bar{L} \cdot \mathbb{E}[1/X^\mathsf{T}\theta_*]$-Lipschitz.*

We stress that Conditions (c) and (d) of Assumption 2 are crucial to establish Lemma 5 but are not relied upon anywhere else in this paper.

**Specification of the algorithm**    At any round $t \in \mathbb{N}$, we bid:

$$b_t = \min(1, \min(1, \max_{\theta \in \mathcal{C}} x_t^\mathsf{T}\theta)/\lambda_t), \tag{3}$$

where $\lambda_t \geq 0$ is the current proxy for $\lambda_*$. We perform a binary search on $\lambda_*$ by repeatedly using the same value of $\lambda_t$ for consecutive rounds forming phases, indexed by $k \in \mathbb{N}$, and by keeping track of an interval, denoted by $I_k = [\underline{\lambda}_k, \bar{\lambda}_k]$. We start with phase $k = 0$ and we initially set $\underline{\lambda}_0 = 0$ and $\bar{\lambda}_0 = 2/r$. The length of the interval is shrunk by half at the end of every phase so that $|I_k| = (2/r)/2^k$ for any $k$. Phase $k$ lasts for $N_k = 3 \cdot 4^k \cdot \ln^2(T)$ rounds during which we set the value of $\lambda_t$ to $\underline{\lambda}_k$. Since $\underline{\lambda}_k$ will be no larger than $\lambda_*$ with high probability, this means that we tend to overbid. Note that there are at most $\bar{k}_T = \inf\{n \in \mathbb{N} \mid \sum_{k=0}^n N_k \geq T\}$ phases overall. The key observation enabling a bisection search approach is that, since the feedback is only partially censored, we can build, at the end of any phase $k$, an empirical estimate of $\phi(\lambda, \mathcal{C})$, which we denote

by $\hat{\phi}_k(\lambda, \mathcal{C})$, for any $\lambda \geq \underline{\lambda}_k$ using all of the $N_k$ samples obtained during phase $k$. The decision rule used to update $I_k$ at the end of phase of $k$ is specified next.

---

**Algorithm 1:** Interval updating procedure at the end of phase $k$

---

**Data**: $\bar{\lambda}_k, \underline{\lambda}_k, \Delta_k = 3\sqrt{2\ln(2T)/N_k}$, and $\hat{\phi}_k(\lambda, \mathcal{C})$ for any $\lambda \geq \underline{\lambda}_k$
**Result**: $\bar{\lambda}_{k+1}$ and $\underline{\lambda}_{k+1}$
$\bar{\gamma}_k = \bar{\lambda}_k, \underline{\gamma}_k = \underline{\lambda}_k$;
**while** $\hat{\phi}_k(\bar{\gamma}_k, \mathcal{C}) > \beta + \Delta_k$ **do**
  | $\bar{\gamma}_k = \bar{\gamma}_k + |I_k|, \underline{\gamma}_k = \underline{\gamma}_k + |I_k|$;
**end**
**if** $\hat{\phi}_k(1/2\bar{\gamma}_k + 1/2\underline{\gamma}_k, \mathcal{C}) \leq \beta + \Delta_k$ **then**
  | $\bar{\lambda}_{k+1} = 1/2\bar{\gamma}_k + 1/2\underline{\gamma}_k, \underline{\lambda}_{k+1} = \underline{\gamma}_k$;
**else**
  | $\bar{\lambda}_{k+1} = \bar{\gamma}_k, \underline{\lambda}_{k+1} = 1/2\bar{\gamma}_k + 1/2\underline{\gamma}_k$;
**end**

---

The splitting decision is trivial when $|\hat{\phi}_k(1/2\bar{\gamma}_k + 1/2\underline{\gamma}_k, \mathcal{C}) - \beta| > \Delta_k$ because we get a clear signal that dominates the stochastic noise to either increase or decrease the current proxy for $\lambda_*$. The tricky situation is when $|\hat{\phi}_k(1/2\bar{\gamma}_k + 1/2\underline{\gamma}_k, \mathcal{C}) - \beta| \leq \Delta_k$, in which case the level of noise is too high to draw any conclusion. In this situation, we always favor a smaller value for $\underline{\lambda}_k$ even if that means shifting the interval upwards later on if we realize that we have made a mistake (which is the purpose of the while loop). This is because we can always recover from underestimating $\lambda_*$ since the feedback is only partially censored. Finally, note that the while loop of Algorithm 1 always ends after a finite number of iterations since $\hat{\phi}_k(2/r, \mathcal{C}) = 0 \leq \beta + \Delta_k$.

**Regret analysis** Just like in Section 2, using concentration inequalities is essential to establish regret bounds but this time we need uniform concentration inequalities. We use the Rademacher complexity approach to concentration inequalities (see, for example, [13] and [15]) to control the deviations of $\hat{\phi}_k(\cdot, \mathcal{C})$ uniformly.

**Lemma 6.** *We have* $\mathbb{P}[\sup_{\lambda \in [\underline{\lambda}_k, 2/r]} |\hat{\phi}_k(\lambda, \mathcal{C}) - \phi(\lambda, \mathcal{C})| \leq \Delta_k] \geq 1 - 1/T$, *for any* $k$.

Next, we bound the number of phases as a function of the time horizon.

**Lemma 7.** *For* $T \geq 3$, *we have* $\bar{k}_T \leq \ln(T+1)$ *and* $4^{\bar{k}_T} \leq \frac{T}{\ln^2(T)} + 1$.

Using Lemma 6, we next show that the stochastic bisection search procedure correctly identifies $\lambda \geq 0$ such that $|\phi(\lambda, \mathcal{C}) - \phi(\lambda_*, \mathcal{C})|$ is small with high probability, which is all we really need to lower bound the rewards accumulated in all rounds given Lemma 4.

**Lemma 8.** *For* $C = \bar{L} \cdot \mathbb{E}[1/X^\intercal \theta_*]$ *and provided that* $T \geq \exp(8r^2/C^2)$, *we have:*

$$\mathbb{P}[\cap_{k=0}^{\bar{k}_T}\{|\hat{\phi}_k(\underline{\lambda}_k, \mathcal{C}) - \phi(\lambda_*, \mathcal{C})| \leq 4C \cdot |I_k|, \ |\phi(\underline{\lambda}_k, \mathcal{C}) - \phi(\lambda_*, \mathcal{C})| \leq 3C \cdot |I_k|\}] \geq 1 - \frac{2\ln^2(T)}{T}.$$

In a last step, we show, using the above result and at the cost of an additive logarithmic term in the regret bound, that we may assume that the advertiser participates in exactly $T$ auctions. This enables us to combine Lemma 4, Lemma 7, and Lemma 8 to establish a distribution-free regret bound.

**Theorem 3.** *Bidding according to (3) incurs a regret* $R_{B,T} = \tilde{O}(\frac{\bar{L} \cdot \mathbb{E}[1/X^\intercal \theta_*]}{r^2} \cdot \sqrt{T} \cdot \ln(T))$.

Observe that Theorem 3 applies in particular when $\theta_*$ is known to the advertiser initially and that the regret bound derived does not depend on $d$.

### 3.2 General case

In this section, we combine the methods developed in Sections 2 and 3.1 to tackle the general case.

**Specification of the algorithm**    At any round $t \in \mathbb{N}$, we bid:

$$b_t = \min(1, \min(1, \max_{\theta \in \mathcal{C}_{\tau_t}} x_t^\intercal \theta)/\lambda_t), \qquad (4)$$

where $\tau_t$ is defined in the last paragraph of Section 2 and $\lambda_t \geq 0$ is specified below. We use the bisection search method developed in Section 3.1 as a subroutine in a master algorithm that also runs in phases. Master phases are indexed by $q = 0, \cdots, Q$ and a new master phase starts whenever $\det(M_t)$ has increased by a factor at least $(1 + A)$ compared to the last time there was an update, for some $A > 0$ of our choosing. By construction, the ellipsoidal confidence set used during the $q$-th master phase is fixed so that we can denote it by $\mathcal{C}_q$. During the $q$-th master phase, we run the bisection search method described in Section 3.1 from scratch for the choice $\mathcal{C} = \mathcal{C}_q$ in order to identify a solution $\lambda_{q,*} \geq 0$ to $\phi(\lambda_{q,*}, \mathcal{C}_q) = \beta$ (or $\lambda_{q,*} = 0$ if no solution exists). Thus, $\lambda_t$ is a proxy for $\lambda_{q,*}$ during the $q$-th master phase. This bisection search lasts for $\bar{k}_q$ phases and stops as soon as we move on to a new master phase. Hence, there are at most $\bar{k}_q \leq \bar{k}_T = \inf\{n \in \mathbb{N} \mid \sum_{k=0}^n N_k \geq T\}$ phases during the $q$-th master phase. We denote by $\underline{\lambda}_{q,k}$ the lower end of the interval used at the $k$-th phase of the bisection search run during the $q$-th master phase.

**Regret analysis**    First we show that there can be at most $O(d \cdot \ln(T \cdot d))$ master phases overall.

**Lemma 9.** *We have $Q \leq \bar{Q} = d \cdot \ln(T \cdot d)/\ln(1 + A)$ almost surely.*

Lemma 9 is important because it implies that the bisection searches run long enough to be able to identify sufficiently good approximate values for $\lambda_{q,*}$. Note that our approach is "doubly" optimistic since both $\underline{\lambda}_{q,k} \leq \lambda_{q,*}$ and $\theta_* \in \mathcal{C}_q$ hold with high probability at any point in time. At a high level, the regret analysis goes as follows. First, just like in Section 3.1, we show, using Lemma 8 and at the cost of an additive logarithmic term in the final regret bound, that we may assume that the advertiser participates in exactly $T$ auctions. Second, we show, using the analysis of Theorem 2, that we may assume that the expected per-round reward obtained during phase $q$ is $\mathbb{E}[\min(1, \max_{\theta \in \mathcal{C}_q} x_t^\intercal \theta)]$ (as opposed to $x_t^\intercal \theta_*$) at any round $t$, up to an additive term of order $\tilde{O}(d \cdot \sqrt{T})$ in the final regret bound. Third, we note that Theorem 3 essentially shows that the expected per-round reward obtained during phase $q$ is $R(\lambda_{q,*}, \mathcal{C}_q)$, up to an additive term of order $\tilde{O}(\sqrt{T})$ in the final regret bound. Finally, what remains to be done is to compare $R(\lambda_{q,*}, \mathcal{C}_q)$ with $R(\lambda_*, \{\theta_*\})$, which is done using Lemmas 2 and 3.

**Theorem 4.** *Bidding according to* (4) *incurs a regret $R_{B,T} = \tilde{O}(d \cdot \frac{\bar{L} \cdot \mathbb{E}[1/X^\intercal \theta^*]}{r^2} \cdot f(A) \cdot \sqrt{T})$, where $f(A) = 1/\ln(1 + A) + \sqrt{1 + A}$.*

## 4    Concluding remark

An interesting direction for future research is to characterize achievable regret bounds, in particular through the derivation of lower bounds on regret. When there is no budget limit and no contextual information, Weed et al. [25] provide a thorough characterization with rates ranging from $\Theta(\ln(T))$ to $\Theta(\sqrt{T})$, depending on whether a margin condition on the underlying distribution is satisfied. These lower bounds carry over to our more general setting and, as a result, the dependence of our regret bounds with respect to $T$ cannot be improved in general. It is however unclear whether the dependence with respect to $d$ is optimal. Based on the lower bounds established by Dani et al. [18] for linear stochastic bandits, a model which is arguably closer to our setting than that of Chu et al. [16] because of the need to estimate the bid multiplier $\lambda_*$, we conjecture that a linear dependence on $d$ is optimal but this calls for more work. Given that the contextual information available in practice is often high-dimensional, developing algorithms that exploit the sparsity of the data in a similar fashion as done in [14] for linear contextual MAB problems is also a promising research direction. In this paper, observing that general BwK problems with contextual information are notoriously hard to solve, we exploit the structure of real-time bidding problems to develop a special-purpose algorithm (a stochastic binary search combined with an ellipsoidal confidence set) to get optimal regret bounds. We believe that the ideas behind this special-purpose algorithm could be adapted for other important applications such as contextual dynamic pricing with limited supply.

**Acknowledgments**

Research funded in part by the Office of Naval Research (ONR) grant N00014-15-1-2083.

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
