[Supplementary Material]

# Supplementary material for "Real-Time Bidding with Side Information"

**Assumption 1.** *The random variables $V$ and $P$ are conditionally independent given $X$. Moreover, there exists $\theta_* \in \mathbb{R}^d$ such that $\mathbb{E}[V \mid X] = X^\mathsf{T}\theta_*$ and $\|\theta_*\|_\infty \leq 1$.*

**Assumption 2.** *(a) $B/T = \beta$ is a constant independent of any other relevant quantities.*

*(b) There exists $r > 0$, known to the advertiser, such that $p_t \geq r$ for all $t \in \mathbb{N}$.*

*(c) We have $\mathbb{E}[1/X^\mathsf{T}\theta_*] < \infty$.*

*(d) The random variable $P$ has a continuous conditional probability density function given the occurrence of the value $x$ of $X$, denoted by $f_x(\cdot)$, that is upper bounded by $\bar{L} < \infty$.*

---

**Algorithm 1:** Interval updating procedure at the end of phase $k$

**Data**: $\bar{\lambda}_k$, $\underline{\lambda}_k$, $\Delta_k = 3\sqrt{2\ln(2T)/N_k}$, and $\hat{\phi}_k(\lambda, \mathcal{C})$ for any $\lambda \geq \underline{\lambda}_k$
**Result**: $\bar{\lambda}_{k+1}$ and $\underline{\lambda}_{k+1}$
$\bar{\gamma}_k = \bar{\lambda}_k, \underline{\gamma}_k = \underline{\lambda}_k$;
**while** $\hat{\phi}_k(\bar{\gamma}_k, \mathcal{C}) > \beta + \Delta_k$ **do**
$\quad \mid \quad \bar{\gamma}_k = \bar{\gamma}_k + |I_k|, \underline{\gamma}_k = \underline{\gamma}_k + |I_k|$;
**end**
**if** $\hat{\phi}_k(1/2\bar{\gamma}_k + 1/2\underline{\gamma}_k, \mathcal{C}) \leq \beta + \Delta_k$ **then**
$\quad \mid \quad \bar{\lambda}_{k+1} = 1/2\bar{\gamma}_k + 1/2\underline{\gamma}_k, \underline{\lambda}_{k+1} = \underline{\gamma}_k$;
**else**
$\quad \mid \quad \bar{\lambda}_{k+1} = \bar{\gamma}_k, \underline{\lambda}_{k+1} = 1/2\bar{\gamma}_k + 1/2\underline{\gamma}_k$;
**end**

---

# A   Proof of Lemma 1

**Lemma 1.** *The optimal non-anticipating strategy is to bid $b_t = x_t^\mathsf{T}\theta_*$ at any time period $t \in \mathbb{N}$ and we have $\mathrm{ER_{OPT}}(T) = \sum_{t=1}^{T} \mathbb{E}[(x_t^\mathsf{T}\theta_* - p_t)_+]$.*

*Proof.* Consider any non-anticipating algorithm. The expected reward obtained at period $t \in \mathbb{N}$ is:

$$
\begin{aligned}
\mathbb{E}[(v_t - p_t) \cdot \mathbb{1}_{b_t \geq p_t}] &= \mathbb{E}[\mathbb{E}[(v_t - p_t) \cdot \mathbb{1}_{b_t \geq p_t} \mid \tilde{\mathcal{F}}_{t-1}]] \\
&= \mathbb{E}[(\mathbb{E}[v_t \mid \tilde{\mathcal{F}}_{t-1}, b_t] - p_t) \cdot \mathbb{1}_{b_t \geq p_t}] \\
&= \mathbb{E}[(x_t^\mathsf{T}\theta_* - p_t) \cdot \mathbb{1}_{b_t \geq p_t}] \\
&\leq \mathbb{E}[(x_t^\mathsf{T}\theta_* - p_t)_+].
\end{aligned}
$$

To derive the second equality, we use the fact that $((x_\tau, v_\tau, p_\tau))_{\tau \in \mathbb{N}}$ is an i.i.d. sequence, that $(v_t, p_t)$ is independent of $b_t$ conditioned on $x_t$ since the algorithm is non-anticipating, and that $v_t$ is independent of $p_t$ conditioned on $x_t$. This shows that:

$$
\mathrm{ER_{OPT}}(T) \leq \sum_{t=1}^{T} \mathbb{E}[(x_t^\mathsf{T}\theta_* - p_t)_+].
$$

Moreover, this last inequality is in fact an equality since bidding $b_t = x_t^\mathsf{T}\theta_*$ at any time period $t \in \mathbb{N}$ yields the expected reward:

$$
\begin{aligned}
\mathbb{E}[(v_t - p_t) \cdot \mathbb{1}_{x_t^\mathsf{T}\theta_* \geq p_t}] &= \mathbb{E}[\mathbb{E}[(v_t - p_t) \cdot \mathbb{1}_{x_t^\mathsf{T}\theta_* \geq p_t} \mid \tilde{\mathcal{F}}_{t-1}]] \\
&= \mathbb{E}[(\mathbb{E}[v_t \mid \tilde{\mathcal{F}}_{t-1}] - p_t) \cdot \mathbb{1}_{x_t^\mathsf{T}\theta_* \geq p_t}] \\
&= \mathbb{E}[(x_t^\mathsf{T}\theta_* - p_t) \cdot \mathbb{1}_{x_t^\mathsf{T}\theta_* \geq p_t}] \\
&= \mathbb{E}[(x_t^\mathsf{T}\theta_* - p_t)_+].
\end{aligned}
$$

$\square$

# B   Proof of Lemma 2

At any round $t$, we bid:

$$
b_t = \max(0, \min(1, \max_{\theta \in \mathcal{C}_t} \theta^\mathsf{T} x_t)) = \max(0, \min(1, \hat{\theta}_t^\mathsf{T} x_t + \delta_T \cdot \sqrt{x_t^\mathsf{T} M_t^{-1} x_t})). \tag{1}
$$

**Lemma 2.** *We have $\mathbb{P}[\theta^* \notin \cap_{t=1}^T \mathcal{C}_t] \leq \frac{1}{T}$.*

*Proof.* This is almost a direct consequence of Theorems 1 and 2 of [1] with the minor change (in their notations): $\eta_t = (v_t - x_t^\mathsf{T}\theta_*) \cdot \mathbb{1}_{b_t \geq p_t}$, $X_t = \mathbb{1}_{b_t \geq p_t} \cdot x_t$, and $Y_t = v_t \cdot \mathbb{1}_{b_t \geq p_t}$. Defining the $\sigma$-algebra $F_t = \sigma(x_1, \cdots, x_{t+1}, p_1, \cdots, p_{t+1}, v_1, \cdots, v_t)$, observe that $X_t$ is $F_{t-1}$-measurable since (1) defines a non-anticipating algorithm, that $\eta_t$ is $F_t$-measurable, and that $\eta_t \in [-1, 1]$ and has mean 0 conditioned on $F_{t-1}$ since $v_t$ is independent of $p_t$ conditioned on $x_t$ with mean $x_t^\mathsf{T}\theta_*$ and since $((x_\tau, v_\tau, p_\tau))_{\tau \in \mathbb{N}}$ is an i.i.d. sequence. This implies that the assumptions of Theorems 1 and 2 of [1] are satisfied with $R = 1$, $V = I_d$, $S = \sqrt{d}$ (since $\|\theta_*\|_\infty \leq 1$), $\delta = 1/T$, and $L = \sqrt{d}$ (since $\|x_t\|_\infty \leq 1$). $\square$

# C   Proof of Theorem 1

**Theorem 1.** *Bidding according to (1) incurs a regret $R_T = \tilde{O}(d \cdot \sqrt{T})$.*

*Proof.* At any time period $t \in \mathbb{N}$, we denote by $\tilde{\theta}_t$ an arbitrary element of $\operatorname{argmax}_{\theta \in \mathcal{C}_t} x_t^\mathsf{T}\theta$ so that $b_t = \max(0, \min(1, x_t^\mathsf{T}\tilde{\theta}_t))$. Using Lemma 1, we have:

$$
\begin{aligned}
R_T &= \sum_{t=1}^T \mathbb{E}[(x_t^\mathsf{T}\theta_* - p_t)_+] - \sum_{t=1}^T \mathbb{E}[(v_t - p_t) \cdot \mathbb{1}_{b_t \geq p_t}] \\
&= \sum_{t=1}^T \mathbb{E}[(x_t^\mathsf{T}\theta_* - p_t)_+] - \sum_{t=1}^T \mathbb{E}[(x_t^\mathsf{T}\theta_* - p_t) \cdot \mathbb{1}_{b_t \geq p_t}].
\end{aligned}
$$

The second equality is derived by conditioning on $\tilde{\mathcal{F}}_{t-1}$ in the same fashion as done in the proof of Lemma 1 since $b_t$ is entirely determined by $\tilde{\mathcal{F}}_{t-1}$, see (1). Observe that:

$$
\begin{aligned}
(x_t^\mathsf{T}\theta_* - p_t)_+ &= (x_t^\mathsf{T}\theta_* - p_t)_+ \cdot \mathbb{1}_{x_t^\mathsf{T}\theta_* \geq p_t > b_t} + (x_t^\mathsf{T}\theta_* - p_t)_+ \cdot \mathbb{1}_{b_t \geq p_t} \\
&\leq (x_t^\mathsf{T}\theta_* - p_t)_+ \cdot \mathbb{1}_{x_t^\mathsf{T}\theta_* > b_t} + (x_t^\mathsf{T}\theta_* - p_t)_+ \cdot \mathbb{1}_{b_t \geq p_t} \\
&\leq \mathbb{1}_{x_t^\mathsf{T}\theta_* > b_t} + (x_t^\mathsf{T}\theta_* - p_t)_+ \cdot \mathbb{1}_{b_t \geq p_t},
\end{aligned}
$$

since $v_t \in [0, 1]$ (which implies that $x_t^\mathsf{T}\theta_* = \mathbb{E}[v_t|x_t] \in [0, 1]$) and $p_t \geq 0$. Plugging this inequality back into the regret bound yields:

$$
\begin{aligned}
R_T &\leq \sum_{t=1}^T \mathbb{P}[x_t^\mathsf{T}\theta_* > b_t] + \mathbb{E}[((x_t^\mathsf{T}\theta_* - p_t)_+ - (x_t^\mathsf{T}\theta_* - p_t)) \cdot \mathbb{1}_{b_t \geq p_t}] \\
&= \sum_{t=1}^T \mathbb{P}[x_t^\mathsf{T}\theta_* > b_t] + \mathbb{E}[(p_t - x_t^\mathsf{T}\theta_*)_+ \cdot \mathbb{1}_{b_t \geq p_t}].
\end{aligned} \tag{2}
$$

Since $x_t^\mathsf{T}\theta_* \in [0,1]$, $x_t^\mathsf{T}\theta_* > b_t$ implies that $x_t^\mathsf{T}\theta_* > \max_{\theta \in \mathcal{C}_t} x_t^\mathsf{T}\theta$ and we conclude that $\theta^* \notin \mathcal{C}_t$. Using Lemma 2, we get:

$$\sum_{t=1}^{T} \mathbb{P}[x_t^\mathsf{T}\theta_* > b_t] \leq \sum_{t=1}^{T} \mathbb{P}[\theta_* \notin \mathcal{C}_t] \leq 1.$$

What remains to be done is to upper bound the second term in the right-hand side of (2). Using Fubini's theorem, we have:

$$\sum_{t=1}^{T} \mathbb{E}[(p_t - x_t^\mathsf{T}\theta_*)_+ \cdot \mathbb{1}_{b_t \geq p_t}] = \int_0^\infty \mathbb{E}[\sum_{t=1}^{T} \mathbb{1}_{p_t - x_t^\mathsf{T}\theta_* \geq u} \cdot \mathbb{1}_{b_t \geq p_t}]du$$

$$\leq \int_0^\infty \mathbb{E}[\sum_{t=1}^{T} \mathbb{1}_{b_t - x_t^\mathsf{T}\theta_* \geq u} \cdot \mathbb{1}_{b_t \geq p_t}]du \qquad (3)$$

$$= \mathbb{E}[\sum_{t=1}^{T} (b_t - x_t^\mathsf{T}\theta_*)_+ \cdot \mathbb{1}_{b_t \geq p_t}].$$

Using Lemma 2, we have that $\theta^* \in \mathcal{C}_t$ (which implies $\left\|\tilde{\theta}_t - \theta^*\right\|_{M_t} \leq 2\delta_T$) for all $t \in \{t, \cdots, T\}$ with probability at least $1 - 1/T$. Using the shorthand $E = \{\theta_* \in \cap_{t=1}^T \mathcal{C}_t\}$, we get:

$$\sum_{t=1}^{T} \mathbb{E}[(b_t - x_t^\mathsf{T}\theta_*)_+ \cdot \mathbb{1}_{b_t \geq p_t}] \leq T \cdot \mathbb{P}[E^\complement] + \mathbb{E}[\mathbb{1}_E \cdot \sum_{t=1}^{T} (b_t - x_t^\mathsf{T}\theta_*)_+ \cdot \mathbb{1}_{b_t \geq p_t}]$$

$$\leq 1 + \mathbb{E}[\mathbb{1}_E \cdot \sum_{t=1}^{T} |x_t^\mathsf{T}\tilde{\theta}_t - x_t^\mathsf{T}\theta_*| \cdot \mathbb{1}_{b_t \geq p_t}]$$

$$\leq 1 + \mathbb{E}[\mathbb{1}_E \cdot \sum_{t=1}^{T} \left\|\mathbb{1}_{b_t \geq p_t} \cdot x_t\right\|_{M_t^{-1}} \cdot \left\|\tilde{\theta}_t - \theta_*\right\|_{M_t}]$$

$$\leq 1 + 2\delta_T \cdot \mathbb{E}[\sum_{t=1}^{T} \left\|\mathbb{1}_{b_t \geq p_t} \cdot x_t\right\|_{M_t^{-1}}]$$

$$\leq 1 + 2\delta_T \cdot \sqrt{d \cdot T \cdot \ln(T)},$$

where we use $b_t \in [0,1]$ and $x_t^\mathsf{T}\theta_* \in [0,1]$ for the first inequality and where the last inequality is derived in Lemma 3 of [2]. □

# D  Proof of Theorem 2

We bid:

$$b_t = \max(0, \min(1, \max_{\theta \in \mathcal{C}_{\tau_t}} \theta^\mathsf{T} x_t)), \qquad (4)$$

at any round $t$, where $\tau_t$ is the last round before round $t$ where the last batch update happened.

**Theorem 2.** *Bidding according to* (4) *at any round $t$ incurs a regret $R_T = \tilde{O}(d \cdot \sqrt{A \cdot T})$.*

*Proof.* At any time period $t \in \mathbb{N}$, we denote by $\tilde{\theta}_t$ an arbitrary element of $\text{argmax}_{\theta \in \mathcal{C}_{\tau_t}} x_t^\mathsf{T}\theta$. The proof is along the same lines as for Theorem 1 except for two inequalities. First, we

now bound the first term in the right-hand side of 2 as follows:

$$\sum_{t=1}^{T} \mathbb{P}[x_t^\intercal \theta_* > b_t] \le \sum_{t=1}^{T} \mathbb{P}[\theta_* \notin \mathcal{C}_{\tau_t}]$$
$$\le \sum_{t=1}^{T} \mathbb{P}[\theta_* \notin \cap_{\tau=1}^{T} \mathcal{C}_\tau]$$
$$\le 1,$$

which leads to the same conclusion. Second, using the shorthand $E = \{\theta_* \in \cap_{t=1}^{T} \mathcal{C}_t\}$, we bound the right-hand side of (3) as follows:

$$\mathbb{E}[\sum_{t=1}^{T} (b_t - x_t^\intercal \theta_*)_+ \cdot \mathbb{1}_{b_t \ge p_t}] \le T \cdot \mathbb{P}[E^{\complement}] + \mathbb{E}[\mathbb{1}_E \cdot \sum_{t=1}^{T} |x_t^\intercal \tilde{\theta}_t - x_t^\intercal \theta_*| \cdot \mathbb{1}_{b_t \ge p_t}]$$
$$\le 1 + 2\sqrt{1+A} \cdot \delta_T \cdot \mathbb{E}[\sum_{t=1}^{T} \|\mathbb{1}_{b_t \ge p_t} \cdot x_t\|_{M_t^{-1}}]$$
$$\le 1 + 2\sqrt{1+A} \cdot \delta_T \cdot \sqrt{d \cdot T \cdot \ln(T)},$$

where the second inequality is a direct consequence of the proof of Theorem 4 in [1] and the last inequality is derived in Lemma 11 of [2] just like for Theorem 1. □

# E   Proof of Lemma 3

**Lemma 3.** *We have* $\mathrm{ER}_{\mathrm{OPT}}(B,T) \le T \cdot R(\lambda_*, \mathcal{C}) + \sqrt{T}/r + 1$, *where* $\lambda_* \ge 0$ *satisfies* $\phi(\lambda_*, \mathcal{C}) = \beta$ *or* $\lambda_* = 0$ *if no such solution exists (i.e. if* $\mathbb{E}[P] < \beta$*) in which case* $\phi(\lambda_*, \mathcal{C}) \le \beta$.

*Proof.* There are two cases depending on whether $\mathbb{E}[P] \ge \beta$ or not.
**Case 1**: $\mathbb{E}[P] < \beta$.
In this case $\lambda_* = 0$ and the total expected reward obtained by any non-anticipating algorithm is:

$$\mathbb{E}[\sum_{t=1}^{\tau^*-1} v_t \cdot \mathbb{1}_{b_t \ge p_t}] \le \mathbb{E}[\sum_{t=1}^{T} v_t]$$
$$= T \cdot \mathbb{E}[V]$$
$$= T \cdot \mathbb{E}[\mathbb{E}[V \mid X]]$$
$$= T \cdot \mathbb{E}[g(X)],$$

which shows that $\mathrm{ER}_{\mathrm{OPT}}(B,T) \le T \cdot R(\lambda_*, \mathcal{C})$.
**Case 2**: $\mathbb{E}[P] \ge \beta$.
The total expected reward obtained by any non-anticipating algorithm can be bounded as

follows:

$$\mathbb{E}[\sum_{t=1}^{\tau^*-1} v_t \cdot \mathbb{1}_{b_t \geq p_t}] \leq \mathbb{E}[\sum_{t=1}^{\tau^*} v_t \cdot \mathbb{1}_{b_t \geq p_t}]$$

$$= \sum_{t=1}^{\infty} \mathbb{E}[I_{\tau^* \geq t} \cdot v_t \cdot \mathbb{1}_{b_t \geq p_t}]$$

$$= \sum_{t=1}^{\infty} \mathbb{E}[I_{\tau^* \geq t} \cdot \mathbb{E}[v_t \mid \tilde{\mathcal{F}}_{t-1}, b_t] \cdot \mathbb{1}_{b_t \geq p_t}]$$

$$= \sum_{t=1}^{\infty} \mathbb{E}[I_{\tau^* \geq t} \cdot \min(1, \max_{\theta \in \mathcal{C}} x_t^\intercal \theta) \cdot \mathbb{1}_{b_t \geq p_t}]$$

$$= \mathbb{E}[\sum_{t=1}^{\tau^*} \min(1, \max_{\theta \in \mathcal{C}} x_t^\intercal \theta) \cdot \mathbb{1}_{b_t \geq p_t}]$$

$$\leq \mathbb{E}[\sum_{t=1}^{\tau^*-1} \min(1, \max_{\theta \in \mathcal{C}} x_t^\intercal \theta) \cdot \mathbb{1}_{b_t \geq p_t}] + 1,$$

where we use the the fact that $((x_t, v_t, p_t))_{t \in \mathbb{N}}$ is an i.i.d. sequence, that $(v_t, p_t)$ is independent of $b_t$ conditioned on $x_t$ since the algorithm is non-anticipating, that $v_t$ is independent of $p_t$ conditioned on $x_t$, and that $\tau^*$ is a stopping time with respect to $((x_t, v_t, p_t))_{t \in \mathbb{N}}$. As a result, up to a constant additive term in the final bound, we just need to bound the performance of any non-anticipating algorithm when the reward obtained at round $t$ is $\min(1, \max_{\theta \in \mathcal{C}} x_t^\intercal \theta) \cdot \mathbb{1}_{b_t \geq p_t}$ as opposed to $v_t \cdot \mathbb{1}_{b_t \geq p_t}$. Observe that, in this setting, the total reward (resp. cost) obtained (resp. incurred) by any non-anticipating algorithm can be written as $\sum_{t=1}^{T} \min(1, \max_{\theta \in \mathcal{C}} x_t^\intercal \theta) \cdot y_t$ (resp. $\sum_{t=1}^{T} p_t \cdot y_t$) where $y_t = \mathbb{1}_{b_t \geq p_t}$ for $t < \tau^*$ and $y_t = 0$ for $t \geq \tau^*$. Remark that $y_t \in [0, 1]$ for all $t \in \{1, \cdots, T\}$ and that, by definition of $\tau^*$, $\sum_{t=1}^{T} p_t \cdot y_t \leq B$. Thus $(y_t)_{t=1,\cdots,T}$ is always a feasible solution to the knapsack problem:

$$\sup_{(\xi_t)_{t=1,\cdots,T}} \quad \sum_{t=1}^{T} \min(1, \max_{\theta \in \mathcal{C}} x_t^\intercal \theta) \cdot \xi_t$$

$$\text{subject to} \quad \sum_{t=1}^{T} p_t \cdot \xi_t \leq B \tag{5}$$

$$\xi_t \in [0, 1], \quad t = 1, \cdots, T.$$

As a consequence, we conclude that the expected total reward obtained by any non-anticipating algorithm is always no larger than the expected optimal value of (5). This reduces the problem of bounding $\mathrm{ER}_{\mathrm{OPT}}(B, T)$ to a stochastic i.i.d. knapsack problem with $T$ items and a knapsack capacity of $B$ when the $t$-th item has value $\min(1, \max_{\theta \in \mathcal{C}} x_t^\intercal \theta)$ and weights $p_t$. The authors of [5] study general stochastic knapsack problems and, adapting their results to our notations, show that the expected optimal value of (5) is equal to $T \cdot R(\lambda_*, \mathcal{C}) + O(1)$, under technical conditions discussed in their paper. For the purpose of being self-contained and in order to relax some of the assumptions made in [5], we derive a weaker bound that will suit our needs. It is well known that filling up the knapsack with the items sorted in descending order of their value-to-weight ratios until the knapsack is full, possibly using a fractional value for the last item, defines an optimal solution to (5), which we denote by $(\xi_t^*)_{t=1,\cdots,T}$. In what follows, we use the shorthand notations $z_t = \min(1, \max_{\theta \in \mathcal{C}} x_t^\intercal \theta)$ and $\xi_t = \mathbb{1}_{z_t \geq \lambda_* \cdot p_t}$, for $t = 1, \cdots, T$. The expected

optimal value of (5) can be bounded as follows:

$$
\begin{aligned}
\mathbb{E}[\sum_{t=1}^{T} z_t \cdot \xi_t^*] &\leq \mathbb{E}[\sum_{t=1}^{T} z_t \cdot \xi_t + \mathbb{1}_{\sum_{t=1}^{T} p_t \cdot \xi_t \leq B} \cdot \sum_{t=1}^{T} z_t \cdot (\xi_t^* - \xi_t)] \\
&= T \cdot R(\lambda_*, \mathcal{C}) + \mathbb{E}[\mathbb{1}_{\sum_{t=1}^{T} p_t \cdot \xi_t \leq B} \cdot \sum_{t=1}^{T} z_t \cdot (\xi_t^* - \xi_t)] \\
&\leq T \cdot R(\lambda_*, \mathcal{C}) + \mathbb{E}[\mathbb{1}_{\sum_{t=1}^{T} p_t \cdot \xi_t \leq B} \cdot \sum_{t=1}^{T} \frac{p_t}{r} \cdot (\xi_t^* - \xi_t)] \\
&\leq T \cdot R(\lambda_*, \mathcal{C}) + \frac{1}{r} \cdot \mathbb{E}[\mathbb{1}_{\sum_{t=1}^{T} p_t \cdot \xi_t \leq B} \cdot (B - \sum_{t=1}^{T} p_t \cdot \xi_t)] \\
&\leq T \cdot R(\lambda_*, \mathcal{C}) + \frac{1}{r} \cdot \mathbb{E}[|\sum_{t=1}^{T} (\beta - p_t \cdot \xi_t)|] \\
&= T \cdot R(\lambda_*, \mathcal{C}) + \frac{1}{r} \cdot \mathbb{E}[|\sum_{t=1}^{T} \epsilon_t \cdot p_t \cdot \xi_t|] \\
&\leq T \cdot R(\lambda_*, \mathcal{C}) + \frac{1}{r} \cdot \mathbb{E}[\sqrt{\sum_{t=1}^{T} |p_t \cdot \xi_t|^2}] \\
&\leq T \cdot R(\lambda_*, \mathcal{C}) + \frac{\sqrt{T}}{r},
\end{aligned}
$$

where $(\epsilon_t)_{t=1,\cdots,T}$ is a collection of $T$ independent Rademacher variables that are jointly independent of $((x_t, p_t))_{t=1,\cdots,T}$. The first inequality is a consequence of the inequality $\xi_t \geq \xi_t^*$ that holds for any $t = 1, \cdots, T$ when $\sum_{t=1}^{T} p_t \cdot \xi_t > B$. This follows from the definition of $\xi_t^*$ and the fact that, in this case, we cannot fit all items of value-to-weight ratio at least $\lambda_*$ in the knapsack when $\sum_{t=1}^{T} p_t \cdot \xi_t > B$. The first equality is a consequence of $\mathbb{E}[z_t \cdot \xi_t] = R(\lambda_*, \mathcal{C})$, which holds by definition of $z_t$ and $\xi_t$. The second inequality is a consequence of $z_t \leq 1 \leq \frac{p_t}{r}$ by Assumption 2 and $\xi_t \leq \xi_t^*$ for any $t = 1, \cdots, T$ when $\sum_{t=1}^{T} p_t \cdot \xi_t \leq B$ given the definition of $\xi_t^*$ and the fact that, in this case, the combination of all items of value-to-weight ratio at least $\lambda_*$ fit in the knapsack. The third inequality is a consequence of $\sum_{t=1}^{T} p_t \cdot \xi_t^* \leq B$ since $(\xi_t^*)_{t=1,\cdots,T}$ is a feasible solution to (5). The fourth inequality is a consequence of $x \cdot \mathbb{1}_{x \geq 0} \leq |x|$ for all $x \in \mathbb{R}$. The second equality is obtained by symmetrization since $p_t \cdot \xi_t$ has mean $\phi(\lambda_*, \mathcal{C}) = \beta$ for any $t = 1, \cdots, T$. The fifth inequality is a direct consequence of the Khintchine inequality. The last inequality results from $p_t \in [0, 1]$ and $\xi_t \in [0, 1]$. This concludes the proof. $\square$

## F    Proof of Lemma 4

**Lemma 4.** *For any $\lambda_1, \lambda_2 \geq 0$, we have: $|R(\lambda_1, \mathcal{C}) - R(\lambda_2, \mathcal{C})| \leq 1/r \cdot |\phi(\lambda_1, \mathcal{C}) - \phi(\lambda_2, \mathcal{C})|$.*

*Proof.* For any $\lambda_1 \geq \lambda_2 \geq 0$, we have:

$$
\begin{aligned}
|R(\lambda_1, \mathcal{C}) - R(\lambda_2, \mathcal{C})| &= \mathbb{E}[g(X) \cdot \mathbb{1}_{g(X)/\lambda_2 \geq P > g(X)/\lambda_1}] \\
&\leq \frac{1}{r} \cdot \mathbb{E}[P \cdot \mathbb{1}_{g(X)/\lambda_2 \geq P > g(X)/\lambda_1}] \\
&= \frac{1}{r} \cdot |\phi(\lambda_1, \mathcal{C}) - \phi(\lambda_2, \mathcal{C})|,
\end{aligned}
$$

where the first inequality is a consequence of $g(X) \in [0, 1]$ and $P \geq r$. $\square$

# G   Proof of Lemma 5

**Lemma 5.** $\phi(\cdot, \mathcal{C})$ *is* $\bar{L} \cdot \mathbb{E}[1/X^{\mathsf{T}}\theta_*]$-*Lipschitz.*

*Proof.* For any $\lambda_2 \geq \lambda_1 > 0$, we have:

$$\begin{aligned}
|\phi(\lambda_2, \mathcal{C}) - \phi(\lambda_1, \mathcal{C})| &= \mathbb{E}[P \cdot \mathbb{1}_{\min(1, g(X)/\lambda_1) \geq P > \min(1, g(X)/\lambda_2)}]\,` \\
&\leq \mathbb{E}[\mathbb{1}_{\min(1, g(X)/\lambda_1) \geq P > \min(1, g(X)/\lambda_2)}] \\
&= \mathbb{E}[\int_{\min(1, g(X)/\lambda_2)}^{\min(1, g(X)/\lambda_1)} f_X(w) \mathrm{d}w] \\
&\leq \bar{L} \cdot \mathbb{E}[\min(1, g(X)/\lambda_1) - \min(1, g(X)/\lambda_2)] \\
&\leq \bar{L} \cdot \mathbb{E}[\frac{1}{g(X)}] \cdot |\lambda_1 - \lambda_2| \\
&\leq \bar{L} \cdot \mathbb{E}[\frac{1}{X^{\mathsf{T}}\theta_*}] \cdot |\lambda_1 - \lambda_2|,
\end{aligned}$$

where the first inequality is obtained using $P \in [0, 1]$, the second inequality is a consequence of Assumption 2, the third inequality actually holds almost surely irrespective of whether $g(X)/\lambda_1 \leq 1$, $g(X)/\lambda_1 > 1$, $g(X)/\lambda_2 \leq 1$, or $g(X)/\lambda_2 > 1$, and the last inequality is obtained using the fact that $\theta_* \in \mathcal{C}$. Also, observe that the last inequality holds even when $\lambda_1 = 0$. □

# H   Proof of Lemma 6

**Lemma 6.** *We have* $\mathbb{P}[\sup_{\lambda \in [\underline{\lambda}_k, 2/r]} |\hat{\phi}_k(\lambda, \mathcal{C}) - \phi(\lambda, \mathcal{C})| \leq \Delta_k] \geq 1 - 1/T$, *for any* $k$.

*Proof.* For any phase $k \in \mathbb{N}$, we denote by $t_k$ the time period at which phase $k$ starts. First note that we can reason conditionally on $\mathcal{F}_{t_{k-1}}$ since $((x_t, v_t, p_t))_{t \in \mathbb{N}}$ is an i.i.d. stochastic process. We use the Rademacher complexity approach to concentration inequalities for empirical processes to derive the result, see, for example, [3] and [4]. Specifically, the class of functions of interest is $\mathcal{F} = \{\ell_\lambda : (x, y) \in [0, 1] \times [r, 1] \to y \cdot \mathbb{1}_{x \geq \lambda \cdot y} \mid \lambda \in [\underline{\lambda}_k, 2/r]\}$. Observe that $\ell_\lambda(x, y) \in [0, 1]$ for any $(x, y) \in [0, 1] \times [r, 1]$ and that $\phi(\lambda, \mathcal{C}) = \mathbb{E}[\ell_\lambda(g(X), P)]$. Moreover, note that $N_k$ samples, denoted by $(\min(1, \max_{\theta \in \mathcal{C}} X_n^{\mathsf{T}}\theta), P_n)_{n=1, \cdots, N_k}$, have been generated according to the same distribution as $(g(X), P)$ in an i.i.d. fashion at the end of phase $k$. Using Theorem 3.2 from [4], we get:

$$\mathbb{P}[\exists \lambda \in [\underline{\lambda}_k, 2/r] \mid \hat{\phi}_k(\lambda, \mathcal{C}) - \phi(\lambda, \mathcal{C})| \geq 2\mathcal{R}_{N_k}(\mathcal{F}) + t \mid \mathcal{F}_{t_{k-1}}] \leq \exp(-2N_k \cdot t^2) \quad \forall t \geq 0, \tag{6}$$

where $\mathcal{R}_{N_k}(\mathcal{F})$ is the Rademacher complexity of $\mathcal{F}$ for $N_k$ samples. What remains to be done is to upper bound this last quantity. By definition, we have, for $N_k$ independent

Rademacher variables $(\epsilon_n)_{n=1,\cdots,N_k}$ that are independent of $(X_n, P_n)_{n=1,\cdots,N_k}$:

$$\mathcal{R}_{N_k}(\mathcal{F}) = \frac{1}{N_k} \cdot \mathbb{E}[\sup_{\lambda \in [\underline{\lambda}_k, 2/r]} |\sum_{n=1}^{N_k} \epsilon_n \cdot \ell_\lambda(\min(1, \max_{\theta \in \mathcal{C}} X_n^\mathsf{T}\theta), P_n)|]$$

$$= \frac{1}{N_k} \cdot \mathbb{E}[\mathbb{E}[\sup_{z \in S((X_n, P_n)_{n=1,\cdots,N_k})} |\sum_{n=1}^{N_k} \epsilon_n \cdot z_n| \mid (X_n, P_n)_{n=1,\cdots,N_k}]]$$

$$\leq \frac{1}{N_k} \cdot \mathbb{E}[\sqrt{2N_k \cdot \ln(2|S((X_n, P_n)_{n=1,\cdots,N_k})|)}]$$

$$\leq \sqrt{\frac{2\ln(2(N_k + 1))}{N_k}}$$

$$\leq \sqrt{\frac{2\ln(2T)}{N_k}},$$

with:

$$S((X_n, P_n)_{n=1,\cdots,N_k}) = \{(P_{f(n)} \cdot \mathbb{1}_{\min(1,\max_{\theta \in \mathcal{C}} X_{f(n)}^\mathsf{T}\theta)/P_{f(n)} \geq \lambda})_{n=1,\cdots,N_k} \mid \lambda \geq 0\},$$

where the permutation $f(\cdot)$ of $\{1, \cdots, N_k\}$ is determined by:

$$\min(1, \max_{\theta \in \mathcal{C}} X_{f(n)}^\mathsf{T}\theta)/P_{f(n)} \leq \cdots \leq \min(1, \max_{\theta \in \mathcal{C}} X_{f(1)}^\mathsf{T}\theta)/P_{f(1)}.$$

The second equality is obtained by reindexing the vector $(z_1, \cdots, z_{N_k})$ according to the mapping $f(\cdot)$ which does not change the inner expectation since $(\epsilon_n)_{n=1,\cdots,N_k}$ is independent of $(X_n, P_n)_{n=1,\cdots,N_k}$. Note that $S((X_n, P_n)_{n=1,\cdots,N_k})$ is always a finite set with cardinality no larger than $N_k + 1$, which yields the second and third inequality using standard bounds on the Rademacher complexity of a finite set, see Theorem 3.3 of [4]. Plugging $t = \sqrt{\frac{2\ln(2T)}{N_k}}$ in (6) and using the definition of $\Delta_k$, we conclude that:

$$\mathbb{P}[\exists \lambda \in [\underline{\lambda}_k, 2/r] \mid \hat{\phi}_k(\lambda, \mathcal{C}) - \phi(\lambda, \mathcal{C})| \geq \Delta_k \mid \mathcal{F}_{t_{k-1}}] \leq \exp(-4\ln(2T)) \leq 1/T,$$

which, in particular, implies that:

$$\mathbb{P}[\sup_{\lambda \in [\underline{\lambda}_k, 2/r]} |\hat{\phi}_k(\lambda, \mathcal{C}) - \phi(\lambda, \mathcal{C})| \leq \Delta_k] \geq 1 - 1/T.$$

$\square$

# I  Proof of Lemma 7

**Lemma 7.** *For $T \geq 3$, we have $\bar{k}_T \leq \ln(T+1)$ and $4^{\bar{k}_T} \leq \frac{T}{\ln^2(T)} + 1$.*

*Proof.* By definition, we have:

$$T \geq \sum_{k=0}^{\bar{k}_T - 1} N_k$$

$$\geq 3 \sum_{k=0}^{\bar{k}_T - 1} 4^k \cdot \ln^2(T)$$

$$\geq (4^{\bar{k}_T} - 1) \cdot \ln^2(T),$$

which implies $4^{\bar{k}_T} \leq T/\ln^2(T) + 1$. Since $\ln^2(T) \geq 1$ for $T \geq 3$, we get $4^{\bar{k}_T} \leq T + 1$. Taking logarithms yields the claim since $\ln(4) \geq 1$. $\square$

# J Proof of Lemma 8

**Lemma 8.** *For $C = \bar{L} \cdot \mathbb{E}[1/X^\intercal \theta_*]$ and provided that $T \geq \exp(8r^2/C^2)$, we have:*

$$\mathbb{P}[\cap_{k=0}^{\bar{k}_T}\{|\hat{\phi}_k(\underline{\lambda}_k,\mathcal{C}) - \phi(\lambda_*,\mathcal{C})| \leq 4C \cdot |I_k|, \; |\phi(\underline{\lambda}_k,\mathcal{C}) - \phi(\lambda_*,\mathcal{C})| \leq 3C \cdot |I_k|\}] \geq 1 - \frac{2\ln^2(T)}{T},$$

*Proof.* To simplify the discussion, we assume that $\mathbb{E}[P] \leq \beta$ so that $\phi(\lambda_*,\mathcal{C}) = \beta$ but the discussion would be almost identical if $\mathbb{E}[P] > \beta$ (in which case $\lambda_* = 0$). For any $k \in \{0,\cdots,\bar{k}_T\}$, we define the event:

$$A_k = \{\lambda_* \geq \underline{\lambda}_k, \; |\hat{\phi}_k(\underline{\lambda}_k,\mathcal{C}) - \beta| \leq 4C \cdot |I_k|, \; |\phi(\underline{\lambda}_{k+1},\mathcal{C}) - \beta| \leq 3C \cdot |I_{k+1}|\}.$$

Using the shorthand $E = \cap_{k=0}^{\bar{k}_T} A_k$, we have:

$$\mathbb{P}[E^{\complement}] \leq \sum_{k=0}^{\bar{k}_T} \mathbb{P}[A_k^{\complement}].$$

Note that we exclude the condition $|\phi(\underline{\lambda}_0,\mathcal{C}) - \beta| \leq 3C \cdot |I_0|$ from the definition of $E$ since this condition is automatically satisfied almost surely given Lemma 5. By induction, we have:

$$\mathbb{P}[A_k^{\complement}] \leq \mathbb{P}[A_0^{\complement}] + \sum_{j=0}^{k-1} \mathbb{P}[A_{j+1}^{\complement} \cap A_j]$$

$$= \sum_{j=0}^{k-1} \mathbb{P}[A_{j+1}^{\complement} \cap A_j]$$

for any $k > 0$ since, by construction, $\lambda_* \in [\underline{\lambda}_0, \bar{\lambda}_0] = [0, 2/r]$ which implies that $\mathbb{P}[A_0^{\complement}] = 0$. Rearranging yields:

$$\mathbb{P}[E^{\complement}] \leq \sum_{k=0}^{\bar{k}_T} (\bar{k}_T - k) \cdot \mathbb{P}[A_{k+1}^{\complement} \cap A_k]$$

$$\leq \sum_{k=0}^{\bar{k}_T} (\bar{k}_T - k) \cdot \mathbb{P}[B_k^{\complement}] + \sum_{k=0}^{\bar{k}_T} (\bar{k}_T - k) \cdot \mathbb{P}[A_{k+1}^{\complement} \cap A_k \cap B_k]$$

$$\leq \frac{1}{T} \cdot \bar{k}_T \cdot (\bar{k}_T + 1) + \sum_{k=0}^{\bar{k}_T} (\bar{k}_T - k) \cdot \mathbb{P}[A_{k+1}^{\complement} \cap A_k \cap B_k]$$

$$\leq \frac{\ln(T+1)^2}{T} + \sum_{k=0}^{\bar{k}_T} (\bar{k}_T - k) \cdot \mathbb{P}[A_{k+1}^{\complement} \cap A_k \cap B_k],$$

where $B_k = \{\sup_{\lambda \in [\underline{\lambda}_k, 2/r]} |\hat{\phi}_k(\lambda,\mathcal{C}) - \phi(\lambda,\mathcal{C})| \leq \Delta_k\}$. We use Lemma 6 to derive the third inequality and Lemma 7 for the last inequality. What remains to be done is to show that the second term in the right-hand side is 0. Consider $k \in \{1,\cdots,\bar{k}_T\}$ and suppose that $A_{k-1}$ and $B_{k-1}$ hold. We show that $A_k$ must hold which will imply that $\mathbb{P}[A_k^{\complement} \cap A_{k-1} \cap B_{k-1}] = 0$. First observe that we have:

$$|\hat{\phi}_k(\underline{\lambda}_k,\mathcal{C}) - \beta| \leq |\hat{\phi}_k(\underline{\lambda}_k,\mathcal{C}) - \phi(\underline{\lambda}_k,\mathcal{C})| + |\phi(\underline{\lambda}_k,\mathcal{C}) - \beta|$$

$$\leq \Delta_k + 3C \cdot |I_k|$$

$$\leq 4C \cdot |I_k|,$$

where we use the fact that $A_{k-1}$ and $B_{k-1}$ hold for the first inequality and the fact that $T \geq \exp(8r^2/C^2)$ for the last inequality. At the end of Algorithm 1 for the $k$-th phase, we end up with an interval $[\underline{\gamma}_k, \bar{\gamma}_k]$ of length $|I_k|$ such that either (i) $\underline{\gamma}_k > \underline{\lambda}_k$ or (ii) $\underline{\gamma}_k = \underline{\lambda}_k$. In situation (i), by definition of the ending criterion of Algorithm 1, we must have $\hat{\phi}_k(\bar{\gamma}_k, \mathcal{C}) \leq \beta + \Delta_k$ and $\hat{\phi}_k(\underline{\gamma}_k, \mathcal{C}) > \beta + \Delta_k$. This last inequality, combined with the fact that $B_{k-1}$ holds, implies that $\phi(\underline{\gamma}_k, \mathcal{C}) > \beta$ and thus we have $\underline{\gamma}_k \leq \lambda_*$. In situation (ii), we automatically have $\underline{\gamma}_k \leq \lambda_*$ since $A_{k-1}$ holds. Moreover, by definition of the ending criterion of Algorithm 1, we must have $\hat{\phi}_k(\bar{\gamma}_k, \mathcal{C}) \leq \beta + \Delta_k$. We conclude that $\underline{\gamma}_k \leq \lambda_*$ and:

$$\hat{\phi}_k(\bar{\gamma}_k, \mathcal{C}) \leq \beta + \Delta_k \tag{7}$$

irrespective of whether (i) or (ii) holds. There are several cases to consider at this point depending on the value of $|\hat{\phi}_k(1/2\underline{\gamma}_k + 1/2\bar{\gamma}_k, \mathcal{C}) - \beta|$. We show that, in any case, we have $\underline{\lambda}_{k+1} \leq \lambda_*$ and $|\phi(\underline{\lambda}_{k+1}, \mathcal{C}) - \beta| \leq 3C \cdot |I_{k+1}|$ which will conclude the proof.

**Case 1**: $\hat{\phi}_k(1/2\underline{\gamma}_k + 1/2\bar{\gamma}_k, \mathcal{C}) < \beta - \Delta_k$.

In this case, we have $\underline{\lambda}_{k+1} = \underline{\gamma}_k \leq \lambda_*$ and $\bar{\lambda}_{k+1} = 1/2\underline{\gamma}_k + 1/2\bar{\gamma}_k$. Using $\hat{\phi}_k(\bar{\lambda}_{k+1}, \mathcal{C}) < \beta - \Delta_k$ along with the fact that $B_{k-1}$ holds, we get $\phi(\bar{\lambda}_{k+1}, \mathcal{C}) < \beta$ which implies that $\lambda_* \in [\underline{\lambda}_{k+1}, \bar{\lambda}_{k+1}]$ and, as a result, $|\phi(\underline{\lambda}_{k+1}, \mathcal{C}) - \beta| = |\phi(\underline{\lambda}_{k+1}, \mathcal{C}) - \phi(\lambda_*, \mathcal{C})| \leq C \cdot |I_{k+1}|$ using Lemma 5.

**Case 2**: $|\hat{\phi}_k(1/2\underline{\gamma}_k + 1/2\bar{\gamma}_k, \mathcal{C}) - \beta| \leq \Delta_k$.

In this case, we have $\underline{\lambda}_{k+1} = \underline{\gamma}_k \leq \lambda_*$ and $\bar{\lambda}_{k+1} = 1/2\underline{\gamma}_k + 1/2\bar{\gamma}_k$. We get:

$$\begin{aligned}
|\phi(\underline{\lambda}_{k+1}, \mathcal{C}) - \beta| &= |\phi(\underline{\lambda}_{k+1}, \mathcal{C}) - \phi(\lambda_*, \mathcal{C})| \\
&\leq |\phi(\underline{\lambda}_{k+1}, \mathcal{C}) - \phi(\bar{\lambda}_{k+1}, \mathcal{C})| \\
&\quad + |\phi(\bar{\lambda}_{k+1}, \mathcal{C}) - \hat{\phi}_k(\bar{\lambda}_{k+1}, \mathcal{C})| \\
&\quad + |\hat{\phi}_k(\bar{\lambda}_{k+1}, \mathcal{C}) - \beta| \\
&\leq C \cdot |I_{k+1}| + \Delta_k + \Delta_k \\
&\leq 3C \cdot |I_{k+1}|,
\end{aligned}$$

where we use Lemma 5, the fact $B_{k-1}$ hold, and $|\hat{\phi}_k(1/2\underline{\gamma}_k + 1/2\bar{\gamma}_k, \mathcal{C}) - \beta| \leq \Delta_k$ for the second inequality while we use $T \geq \exp(8r^2/C^2)$ for the last inequality.

**Case 3**: $\hat{\phi}_k(1/2\underline{\gamma}_k + 1/2\bar{\gamma}_k, \mathcal{C}) > \beta + \Delta_k$.

In this case, $\underline{\lambda}_{k+1} = 1/2\underline{\gamma}_k + 1/2\bar{\gamma}_k$ and $\bar{\lambda}_{k+1} = \bar{\gamma}_k$. Since $B_{k-1}$ holds, we get $\phi(\underline{\lambda}_{k+1}, \mathcal{C}) > \beta$ and thus $\underline{\lambda}_{k+1} \leq \lambda_*$. Using (7), we have either (a) $\hat{\phi}_k(\bar{\gamma}_k, \mathcal{C}) < \beta - \Delta_k$ or (b) $|\hat{\phi}_k(\bar{\gamma}_k, \mathcal{C}) - \beta| \leq \Delta_k$. If (a) is true then, since $B_{k-1}$ holds, it must be that $\phi(\bar{\lambda}_{k+1}, \mathcal{C}) < \beta$ and thus we get $\lambda_* \in [\underline{\lambda}_{k+1}, \bar{\lambda}_{k+1}]$ which implies that $|\phi(\underline{\lambda}_{k+1}, \mathcal{C}) - \beta| = |\phi(\underline{\lambda}_{k+1}, \mathcal{C}) - \phi(\lambda_*, \mathcal{C})| \leq C \cdot |I_{k+1}|$ using Lemma 5. If (b) is true then we have:

$$\begin{aligned}
|\phi(\underline{\lambda}_{k+1}, \mathcal{C}) - \beta| &= |\phi(\underline{\lambda}_{k+1}, \mathcal{C}) - \phi(\lambda_*, \mathcal{C})| \\
&\leq |\phi(\underline{\lambda}_{k+1}, \mathcal{C}) - \phi(\bar{\lambda}_{k+1}, \mathcal{C})| \\
&\quad + |\phi(\bar{\lambda}_{k+1}, \mathcal{C}) - \hat{\phi}_k(\bar{\lambda}_{k+1}, \mathcal{C})| \\
&\quad + |\hat{\phi}_k(\bar{\lambda}_{k+1}, \mathcal{C}) - \beta| \\
&\leq C \cdot |I_{k+1}| + \Delta_k + \Delta_k \\
&\leq 3C \cdot |I_{k+1}|.
\end{aligned}$$

where we use (7), the fact that $B_{k-1}$ holds, and (b) for the second inequality while we use $T \geq \exp(8r^2/C^2)$ for the last inequality. $\square$

# K  Proof of Theorem 3

At any round $t \in \mathbb{N}$, we bid:

$$b_t = \min(1, \min(1, \max_{\theta \in \mathcal{C}} x_t^\mathsf{T}\theta)/\lambda_t). \tag{8}$$

**Theorem 3.** *Bidding according to* (8) *incurs a regret* $R_{B,T} = \tilde{O}(\frac{\bar{L} \cdot \mathbb{E}[1/X^\mathsf{T}\theta^*]}{r^2} \cdot \sqrt{T} \cdot \ln(T))$.

*Proof.* For any phase $k \in \mathbb{N}$, we denote by $t_k$ the time period at which phase $k$ starts. Using Lemma 3, we have:

$$R_{B,T} \leq T \cdot R(\lambda_*, \mathcal{C}) - \mathbb{E}[\sum_{t=1}^{\tau^*-1} v_t \cdot \mathbb{1}_{b_t \geq p_t}] + O(\frac{\sqrt{T}}{r})$$

$$= T \cdot R(\lambda_*, \mathcal{C}) - \mathbb{E}[\sum_{t=1}^{\tau^*} v_t \cdot \mathbb{1}_{b_t \geq p_t}] + O(\frac{\sqrt{T}}{r}).$$

Since $\tau^*$ is a stopping time with respect to the sequence $((x_t, v_t, p_t))_{t \in \mathbb{N}}$ and since $b_t = \min(1, \min(1, \max_{\theta \in \mathcal{C}} x_t^\mathsf{T}\theta)/\lambda_t)$ is $\tilde{\mathcal{F}}_{t-1}$-measurable, we have:

$$\mathbb{E}[\sum_{t=1}^{\tau^*} v_t \cdot \mathbb{1}_{b_t \geq p_t}] = \sum_{t=1}^{\infty} \mathbb{E}[\mathbb{1}_{\tau^* \geq t} \cdot \mathbb{E}[v_t \mid \tilde{\mathcal{F}}_{t-1}] \cdot \mathbb{1}_{b_t \geq p_t}]$$

$$= \sum_{t=1}^{\infty} \mathbb{E}[\mathbb{1}_{\tau^* \geq t} \cdot \min(1, \max_{\theta \in \mathcal{C}} x_t^\mathsf{T}\theta) \cdot \mathbb{1}_{b_t \geq p_t}]$$

$$= \mathbb{E}[\sum_{t=1}^{\tau^*} \min(1, \max_{\theta \in \mathcal{C}} x_t^\mathsf{T}\theta) \cdot \mathbb{1}_{b_t \geq p_t}]$$

$$= \sum_{t=1}^{T} \mathbb{E}[\min(1, \max_{\theta \in \mathcal{C}} x_t^\mathsf{T}\theta) \cdot \mathbb{1}_{b_t \geq p_t}] - \mathbb{E}[\sum_{t=\tau^*+1}^{T} \min(1, \max_{\theta \in \mathcal{C}} x_t^\mathsf{T}\theta) \cdot \mathbb{1}_{b_t \geq p_t}]$$

$$\geq \sum_{t=1}^{T} \mathbb{E}[\min(1, \max_{\theta \in \mathcal{C}} x_t^\mathsf{T}\theta) \cdot \mathbb{1}_{b_t \geq p_t}] - \frac{1}{r} \cdot \mathbb{E}[\sum_{t=\tau^*+1}^{T} p_t \cdot \mathbb{1}_{b_t \geq p_t}],$$

where we use $\min(1, \max_{\theta \in \mathcal{C}} x_t^\mathsf{T}\theta) \leq 1$, $p_t \geq r$, and the fact that $v_t$ is independent of $p_t$ conditioned on $x_t$. Observe that:

$$\sum_{t=\tau^*+1}^{T} p_t \cdot \mathbb{1}_{b_t \geq p_t} = 0 \leq (\sum_{t=1}^{T} p_t \cdot \mathbb{1}_{b_t \geq p_t} - B)_+,$$

if $\tau^* = T + 1$ while:

$$\sum_{t=\tau^*+1}^{T} p_t \cdot \mathbb{1}_{b_t \geq p_t} \leq \sum_{t=\tau^*+1}^{T} p_t \cdot \mathbb{1}_{b_t \geq p_t} + \sum_{t=1}^{\tau^*} p_t \cdot \mathbb{1}_{b_t \geq p_t} - B$$

$$\leq (\sum_{t=1}^{T} p_t \cdot \mathbb{1}_{b_t \geq p_t} - B)_+$$

if $\tau^* < T + 1$ since, in this case, we have run out of resources before round $T$, i.e. $\sum_{t=1}^{\tau^*} p_t \cdot \mathbb{1}_{b_t \geq p_t} \geq B$. We derive:

$$R_{B,T} \leq T \cdot R(\lambda_*, \mathcal{C}) - \sum_{t=1}^{T} \mathbb{E}[\min(1, \max_{\theta \in \mathcal{C}} x_t^\mathsf{T}\theta) \cdot \mathbb{1}_{b_t \geq p_t}] + \frac{1}{r} \cdot \mathbb{E}[(\sum_{t=1}^{T} p_t \cdot \mathbb{1}_{b_t \geq p_t} - B)_+]$$

$$+ O(\frac{\sqrt{T}}{r}). \tag{9}$$

We bound the two terms appearing in the right-hand side of (9) separately starting with the first one. Using the shorthand notation:

$$E = \cap_{k=0}^{\bar{k}_T}\{|\hat{\phi}_k(\lambda_k, \mathcal{C}) - \phi(\lambda_*, \mathcal{C})| \leq 4C \cdot |I_k|, \ |\phi(\lambda_k, \mathcal{C}) - \phi(\lambda_*, \mathcal{C})| \leq 3C \cdot |I_k|\},$$

where $C = \bar{L} \cdot \mathbb{E}[\frac{1}{X^\top \theta_*}]$, we have:

$$T \cdot R(\lambda_*, \mathcal{C}) - \sum_{t=1}^{T}\mathbb{E}[\min(1, \max_{\theta \in \mathcal{C}} x_t^\top \theta) \cdot \mathbb{1}_{b_t \geq p_t}]$$

$$= \sum_{t=1}^{T}\{R(\lambda_*, \mathcal{C}) - \mathbb{E}[\min(1, \max_{\theta \in \mathcal{C}} x_t^\top \theta) \cdot \mathbb{1}_{\min(1, \max_{\theta \in \mathcal{C}} x_t^\top \theta) \geq \lambda_t \cdot p_t}]\}$$

$$= \sum_{t=1}^{T}\{R(\lambda_*, \mathcal{C}) - \mathbb{E}[\mathbb{E}[\min(1, \max_{\theta \in \mathcal{C}} x_t^\top \theta) \cdot \mathbb{1}_{\min(1, \max_{\theta \in \mathcal{C}} x_t^\top \theta) \geq \lambda_t \cdot p_t} \mid \mathcal{F}_{t-1}]]\}$$

$$= \sum_{t=1}^{T}\{R(\lambda_*, \mathcal{C}) - \mathbb{E}[R(\lambda_t, \mathcal{C})]\}$$

$$\leq \sum_{t=1}^{T}\mathbb{E}[|R(\lambda_*, \mathcal{C}) - R(\lambda_t, \mathcal{C})|]$$

$$\leq \frac{1}{r} \cdot \sum_{t=1}^{T}\mathbb{E}[|\phi(\lambda_*, \mathcal{C}) - \phi(\lambda_t, \mathcal{C})|]$$

$$\leq \frac{1}{r} \cdot \sum_{k=0}^{\bar{k}_T}N_k \cdot \mathbb{E}[|\phi(\lambda_*, \mathcal{C}) - \phi(\lambda_k, \mathcal{C})|]$$

$$\leq \frac{T}{r} \cdot \mathbb{P}[E^\complement] + \frac{1}{r} \cdot \sum_{k=0}^{\bar{k}_T}N_k \cdot \mathbb{E}[|\phi(\lambda_*, \mathcal{C}) - \phi(\lambda_k, \mathcal{C})| \cdot \mathbb{1}_E]$$

$$\leq 2\frac{\ln^2(T)}{r} + \frac{3C}{r} \cdot \sum_{k=0}^{\bar{k}_T}N_k \cdot |I_k|$$

$$\leq 2\frac{\ln^2(T)}{r} + \frac{18C}{r^2} \cdot \sum_{k=0}^{\bar{k}_T}2^k \cdot \ln^2(T)$$

$$\leq 2\frac{\ln^2(T)}{r} + \frac{36C}{r^2} \cdot \sqrt{T} \cdot \ln(T).$$

To derive the third equality we use the fact that $\lambda_t$ is $\mathcal{F}_{t-1}$-measurable. For the second inequality, we use Lemma 4 For the fourth inequality, we use $\phi(\lambda_k, \mathcal{C}), \beta \in [0,1]$. We use Lemma 8 to derive the fifth inequality while we use Lemma 7 for the last one. We can

now focus on the second term appearing in the right-hand side of (9):

$$\mathbb{E}[(\sum_{t=1}^{T} p_t \cdot \mathbb{1}_{b_t \geq p_t} - B)_+] = \mathbb{E}[(\sum_{t=1}^{T} p_t \cdot \mathbb{1}_{\min(1,\max_{\theta \in \mathcal{C}} x_t^\intercal \theta) \geq \lambda_t \cdot p_t} - B)_+]$$

$$\leq \sum_{k=0}^{\bar{k}_T - 1} \mathbb{E}[(\sum_{t=t_k}^{t_{k+1}-1} p_t \cdot \mathbb{1}_{\min(1,\max_{\theta \in \mathcal{C}} x_t^\intercal \theta) \geq \underline{\lambda}_k \cdot p_t} - N_k \cdot \beta)_+]$$

$$+ \mathbb{E}[(\sum_{t=t_{\bar{k}_T}}^{T} p_t \cdot \mathbb{1}_{\min(1,\max_{\theta \in \mathcal{C}} x_t^\intercal \theta) \geq \lambda_{\bar{k}_T} \cdot p_t} - (T - \sum_{k=0}^{\bar{k}_T - 1} N_k) \cdot \beta)_+]$$

$$\leq \sum_{k=0}^{\bar{k}_T - 1} N_k \cdot \mathbb{E}[(\hat{\phi}_k(\underline{\lambda}_k, \mathcal{C}) - \phi(\lambda_*, \mathcal{C}))_+]$$

$$+ (T - \sum_{k=0}^{\bar{k}_T - 1} N_k) \cdot \mathbb{E}[(\hat{\phi}_{\bar{k}_T}(\underline{\lambda}_{\bar{k}_T}, \mathcal{C}) - \phi(\lambda_*, \mathcal{C}))_+]$$

$$\leq T \cdot \mathbb{P}[E^{\complement}] + \sum_{k=0}^{\bar{k}_T} N_k \cdot \mathbb{E}[|\hat{\phi}_k(\underline{\lambda}_k, \mathcal{C}) - \phi(\lambda_*, \mathcal{C})| \cdot \mathbb{1}_E]$$

$$\leq 2\ln^2(T) + 4C \cdot \sum_{k=0}^{\bar{k}_T} N_k \cdot |I_k|$$

$$\leq 2\ln^2(T) + \frac{24C}{r} \cdot \sum_{k=0}^{\bar{k}_T} 2^k \cdot \ln^2(T)$$

$$\leq 2\ln^2(T) + \frac{48C}{r} \cdot \sqrt{T} \cdot \ln(T).$$

To derive the second inequality, we use $\beta \geq \phi(\lambda_*, \mathcal{C})$. To derive the third inequality, we use $\hat{\phi}_k(\lambda_k, \mathcal{C}), \phi(\lambda_*, \mathcal{C}) \in [0, 1]$ and $N_{\bar{k}_T} \geq T - \sum_{k=0}^{\bar{k}_T - 1} N_k$. We use Lemma 8 to derive the fourth inequality while we use Lemma 7 for the last one. $\qquad\square$

## L    Proof of Lemma 9

**Lemma 9.** *We have* $Q \leq \bar{Q} = d \cdot \ln(T \cdot d) / \ln(1 + A)$ *almost surely.*

*Proof.* We have:

$$\det(M_0) \cdot (1 + A)^Q \leq \det(M_T)$$
$$\leq \det((T \cdot d) I_d)$$
$$= (T \cdot d)^d,$$

by definition of $Q$. The second inequality is obtained using $\|x_t\|_\infty \leq 1$ (which implies that $dI_d - x_t x_t^\intercal$ is positive semidefinite) and the fact that $\det(B + C) \geq \det(B)$ for positive semidefinite matrices $B$ and $C$. Taking logarithms yields the claim. $\qquad\square$

## M    Proof of Theorem 4

At any round $t \in \mathbb{N}$, we bid:

$$b_t = \min(1, \min(1, \max_{\theta \in \mathcal{C}_{\tau_t}} x_t^\intercal \theta) / \lambda_t). \tag{10}$$

**Theorem 4.** *Bidding according to (10) incurs a regret* $R_{B,T} = \tilde{O}(d \cdot \frac{\bar{L} \cdot \mathbb{E}[1/X^{\mathsf{T}}\theta^*]}{r^2} \cdot f(A) \cdot \sqrt{T})$, *where* $f(A) = 1/\ln(1+A) + \sqrt{1+A}$.

*Proof.* For any master phase $q \in \{0, \cdots, \bar{Q}\}$, we denote by $t_q \in \mathbb{N}$ the round at which phase $q$ starts. For any master phase $q \in \{0, \cdots, \bar{Q}\}$, any phase $k \in \{0, \cdots, \bar{k}_T\}$, and any $\lambda \geq 0$, we denote by $\hat{\phi}_{q,k}(\lambda, \mathcal{C}_q)$ the empirical estimate of $\phi(\lambda, \mathcal{C}_q)$ using all $N_k$ samples obtained during the $k$-th phase of the binary search that runs during the $q$-th master phase. We also use the shorthand notations $E = \{\theta_* \in \cap_{t=1}^T \mathcal{C}_t\}$ and:

$$E_q = \cap_{k=0}^{\bar{k}_T} \{|\hat{\phi}_{q,k}(\lambda_{q,k}, \mathcal{C}_q) - \phi(\lambda_{q,*}, \mathcal{C}_q)| \leq 4C \cdot |I_k|, \ |\phi(\lambda_{q,k}, \mathcal{C}_q) - \phi(\lambda_{q,*}, \mathcal{C}_q)| \leq 3C \cdot |I_k|\},$$

for any $q \in \{0, \cdots, \bar{Q}\}$. Using the same analysis as in the proof of Theorem 3 with $\mathcal{C} = \{\theta_*\}$ (see (9)), we derive:

$$R_{B,T} \leq T \cdot R(\lambda_*, \{\theta_*\}) - \sum_{t=1}^T \mathbb{E}[x_t^{\mathsf{T}}\theta_* \cdot \mathbb{1}_{b_t \geq p_t}] + \frac{1}{r} \cdot \mathbb{E}[(\sum_{t=1}^T p_t \cdot \mathbb{1}_{b_t \geq p_t} - B)_+] + O(\frac{\sqrt{T}}{r}). \tag{11}$$

We first study the third term in (11). Observe that, along the same lines as what is done in the proof of Theorem 3, we have:

$$\mathbb{E}[(\sum_{t=1}^T p_t \cdot \mathbb{1}_{b_t \geq p_t} - B)_+] \leq \mathbb{E}[(\sum_{t=1}^T p_t \cdot \mathbb{1}_{b_t \geq p_t} - B)_+ \cdot \mathbb{1}_E] + T \cdot \mathbb{P}[E]$$

$$\leq \mathbb{E}[\sum_{q=0}^{Q} \sum_{k=0}^{\bar{k}_q} N_k \cdot |\hat{\phi}_{q,k}(\lambda_{q,k}, \mathcal{C}_q) - \phi(\lambda_{q,*}, \mathcal{C}_q)| \cdot \mathbb{1}_E] + 1$$

$$\leq \mathbb{E}[\sum_{q=0}^{Q} \sum_{k=0}^{\bar{k}_q} 4N_k \cdot C \cdot |I_k|] + T \cdot \sum_{q=0}^{\bar{Q}} \mathbb{P}[E_q^{\complement} \cap E] + O(1)$$

$$\leq \frac{24C}{r} \cdot \sum_{q=0}^{\bar{Q}} \sum_{k=0}^{\bar{k}_T} 2^k \cdot \ln^2(T) + 2\ln^2(T) \cdot (\bar{Q}+1) + O(1)$$

$$\leq \frac{48C}{r} \cdot \sqrt{T} \cdot \ln(T) \cdot (\bar{Q}+1) + 2\ln^2(T) \cdot (\bar{Q}+1) + O(1)$$

$$= O(\frac{d \cdot C}{r \cdot \ln(1+A)} \cdot \sqrt{T} \cdot \ln^2(T \cdot d))$$

$$= \tilde{O}(\frac{d \cdot C}{r \cdot \ln(1+A)} \cdot \sqrt{T}).$$

We use the same analysis as in the proof of Theorem 3 along with Lemma 2 to derive the second inequality. We use Lemma 8 for the fourth inequality and we use Lemma 9 to get the final asymptotic bound. We move on to study the second term in (11). Denoting by $\tilde{\theta}_t$ an arbitrary element of $\text{argmax}_{\theta \in \mathcal{C}_{\tau_t}} x_t^{\mathsf{T}}\theta$, we have:

$$\sum_{t=1}^T \mathbb{E}[x_t^{\mathsf{T}}\theta_* \cdot \mathbb{1}_{b_t \geq p_t}] = \sum_{t=1}^T \mathbb{E}[\min(1, \max_{\theta \in \mathcal{C}_{\tau_t}} x_t^{\mathsf{T}}\theta) \cdot \mathbb{1}_{b_t \geq p_t}]$$

$$- \mathbb{E}[\sum_{t=1}^T (\min(1, \max_{\theta \in \mathcal{C}_{\tau_t}} x_t^{\mathsf{T}}\theta) - x_t^{\mathsf{T}}\theta_*) \cdot \mathbb{1}_{b_t \geq p_t}]$$

$$\geq \sum_{t=1}^T \mathbb{E}[R(\lambda_t, \mathcal{C}_{\tau_t})] - \mathbb{E}[\sum_{t=1}^T |x_t^{\mathsf{T}}\tilde{\theta}_t - x_t^{\mathsf{T}}\theta_*| \cdot \mathbb{1}_{b_t \geq p_t}]$$

$$\geq \sum_{t=1}^T \mathbb{E}[R(\lambda_t, \mathcal{C}_{\tau_t})] + \tilde{O}(d \cdot \sqrt{A \cdot T}),$$

where the last inequality is obtained in the proof of Theorem 2. Hence, what remains to be done to get the regret bound is to upper bound:

$$T \cdot R(\lambda_*, \{\theta_*\}) - \sum_{t=1}^{T} \mathbb{E}[R(\lambda_t, \mathcal{C}_{\tau_t})].$$

First note that:

$$\mathbb{E}[|\sum_{t=1}^{T} R(\lambda_t, \mathcal{C}_{\tau_t}) - \sum_{q=0}^{Q} (t_{q+1} - t_q) \cdot R(\lambda_{q,*}, \mathcal{C}_q)|]$$

$$\leq \mathbb{E}[\sum_{q=0}^{Q} \sum_{k=0}^{\bar{k}_q} N_k \cdot |R(\underline{\lambda}_{q,k}, \mathcal{C}_q) - R(\lambda_{q,*}, \mathcal{C}_q)|]$$

$$\leq \frac{1}{r} \cdot \mathbb{E}[\sum_{q=0}^{Q} \sum_{k=0}^{\bar{k}_q} N_k \cdot |\phi(\underline{\lambda}_{q,k}, \mathcal{C}_q) - \phi(\lambda_{q,*}, \mathcal{C}_q)|]$$

$$\leq \frac{1}{r} \cdot (T \cdot \mathbb{P}[E] + \sum_{q=0}^{\bar{Q}} T \cdot \mathbb{P}[E_q^{\complement} \cap E] + \sum_{q=0}^{\bar{Q}} \sum_{k=0}^{\bar{k}_T} 3N_k \cdot C \cdot |I_k|)$$

$$\leq \frac{1}{r} \cdot (1 + 2\ln^2(T) \cdot (\bar{Q} + 1) + \frac{48C}{r} \cdot \sqrt{T} \cdot \ln(T) \cdot (\bar{Q} + 1))$$

$$= \tilde{O}(\frac{d \cdot C}{r^2 \cdot \ln(1 + A)} \cdot \sqrt{T}).$$

We derive the second inequality using Lemma 4 We derive the fourth inequality using Lemma 2 and Lemma 8 in the same fashion as done for the third term in (11). We conclude that all that is left to be done is to upper bound:

$$T \cdot R(\lambda_*, \{\theta_*\}) - \mathbb{E}[\sum_{q=0}^{Q} (t_{q+1} - t_q) \cdot R(\lambda_{q,*}, \mathcal{C}_q)],$$

which we do next. Using Lemma 3, observe that, conditioned on $\mathcal{F}_{t_q-1}$ and assuming that $\theta_* \in \mathcal{C}_q$, $R(\lambda_{q,*}, \mathcal{C}_q)$ is almost surely larger than $\mathrm{ER}_{\mathrm{OPT}}(B, T)/T - 1/(r \cdot \sqrt{T}) - 1/T$ by definition of $\lambda_{q,*}$ when $\mathcal{C} = \mathcal{C}_q$. Note that bidding $\tilde{b}_t = \min(x_t^\mathsf{T}\theta_*/\lambda_*, 1)$ at any time period $t$ is a valid algorithm for this problem that yields an expected total reward:

$$\mathbb{E}[\sum_{t=1}^{\tau^*} v_t \cdot \mathbb{1}_{\tilde{b}_t \geq p_t}]$$

$$\geq T \cdot \mathbb{E}[\min(1, \max_{\theta \in \mathcal{C}_q} X^\mathsf{T}\theta) \cdot \mathbb{1}_{X^\mathsf{T}\theta_* \geq \lambda_* \cdot P}] - \frac{1}{r} \cdot \mathbb{E}[(\sum_{t=1}^{T} p_t \cdot \mathbb{1}_{\tilde{b}_t \geq p_t} - B)_+]$$

$$\geq T \cdot \mathbb{E}[\min(1, \max_{\theta \in \mathcal{C}_q} X^\mathsf{T}\theta) \cdot \mathbb{1}_{X^\mathsf{T}\theta_* \geq \lambda_* \cdot P}] - \frac{1}{r} \cdot \mathbb{E}[|\sum_{t=1}^{T} p_t \cdot \mathbb{1}_{\tilde{b}_t \geq p_t} - T \cdot \phi(\lambda_*, \{\theta_*\})|]$$

$$\geq T \cdot \mathbb{E}[\min(1, \max_{\theta \in \mathcal{C}_q} X^\mathsf{T}\theta) \cdot \mathbb{1}_{X^\mathsf{T}\theta_* \geq \lambda_* \cdot P}] - \frac{\sqrt{T}}{r},$$

where the expectations are all conditioned on $\mathcal{F}_{t_q-1}$ and the inequalities hold almost surely. The first inequality is derived in the same fashion as done in the proof of Theorem 3 to derive (9). The second inequality is a consequence of $B = \beta \cdot T$ and $\phi(\lambda_*, \{\theta_*\}) \leq \beta$. The third inequality is obtained with Khintchine's inequality (by symmetrization) since $p_t \in [0, 1]$ and $(p_t \cdot \mathbb{1}_{\tilde{b}_t \geq p_t})_{t \in \mathbb{N}}$ is an i.i.d. stochastic process with mean $\phi(\lambda_*, \{\theta_*\})$. We

conclude that:

$$R(\lambda_{q,*}, \mathcal{C}_q) \geq \mathbb{E}[\min(1, \max_{\theta \in \mathcal{C}_q} X^\mathsf{T}\theta) \cdot \mathbb{1}_{X^\mathsf{T}\theta_* \geq \lambda_* \cdot P} | \mathcal{F}_{t_q-1}] - \frac{2}{r \cdot \sqrt{T}} - \frac{1}{T}$$

$$\geq \mathbb{E}[X^\mathsf{T}\theta_* \cdot \mathbb{1}_{X^\mathsf{T}\theta_* \geq \lambda_* \cdot P} | \mathcal{F}_{t_q-1}] - \frac{2}{r \cdot \sqrt{T}} - \frac{1}{T}$$

$$= R(\lambda_*, \{\theta_*\}) - \frac{2}{r \cdot \sqrt{T}} - \frac{1}{T}$$

almost surely as long as $\theta_* \in \mathcal{C}_q$. This implies that:

$$T \cdot R(\lambda_*, \{\theta_*\}) - \mathbb{E}[\sum_{q=0}^{Q} (t_{q+1} - t_q) \cdot R(\lambda_{q,*}, \mathcal{C}_q)]$$

$$= \mathbb{E}[\sum_{q=0}^{Q} (t_{q+1} - t_q) \cdot (R(\lambda_*, \{\theta_*\}) - R(\lambda_{q,*}, \mathcal{C}_q))]$$

$$\leq \mathbb{E}[\sum_{q=0}^{Q} (t_{q+1} - t_q) \cdot (\mathbb{1}_E + \frac{2}{r \cdot \sqrt{T}} + \frac{1}{T})]$$

$$\leq T \cdot \mathbb{P}[E] + 2\frac{\sqrt{T}}{r} + 1$$

$$= O(\frac{\sqrt{T}}{r}),$$

where we use Lemma 2 for the last step. This concludes the proof. $\qquad\square$