[Reviews · NeurIPS 2017]

Reviewer 1



The paper analyzes the problem of a bidder participating in multiple auctions with a budget B and a horizon T. The objective is to optimize the bids over time, which is modeled as a contextual bandit problem. The paper develops an algorithm with regret of order O(d\sqrt{T}) when there is no budget or when B scales with T. The paper needs to discuss the current literature on learning in Auctions (with or without budgets). For example, see Learning in Repeated Auctions with Budgets: Regret Minimization and Equilibrium. Balseiro and Gur, 2017 and earlier references therein. For example, the bid shading parameter $\lambda$ that appears is in line with what has appeared in the earlier literature. The paper would also benefit from better comparing to other papers that deal with similar knapsack constraints: see, e.g. Close the Gaps: A Learning-while-Doing Algorithm for Single-Product Revenue Management Problems Zizhuo Wang Shiming Deng and Yinyu Ye. Operations Research, 2014 The novelty is the introduction of context in conjunction with budget. As a result, the paper would benefit from highlighting more why the conjunction of context and budget is challenging? Where are the exact novel technical ideas needed for this compared to existing ideas in the contextual bandit literature and in the learning with knapsack constraints'' literatures? Relatedly, the lack of lower bound that captures the dependence on d and T significantly weakens the submission. A sharp lower bound would allow to much better appreciate the quality of the algorithms proposed.

Reviewer 2



The paper studies the problem of repeated bidding in online ad auctions where the goal is to maximize the accumulated reward over a fixed horizon and in the presence of budget constraint. It is also assumed that a context vector is available at each auctions which summarizes the specific information about the targeted person and the website. The expected value of an ad is considered to be a linear function of this context vector. This is a generalization of the setting considered in Weed et al (2016) in two directions: first having a budget constraint and second the availability of the side information. The paper proposes a bidding strategy in such a setting and provide a regret bound for it. While the paper technically sounds and is well-written, here are a few comments on the formulated problem. 1. It has been assumed in lines 51-53 that p_t is an iid process. This seems to be far from reality as p_t is the maximum bid of the competitors each of which might be a learning agent (just like us). In this case, competitors select their bids based on his observed history. This suggests that competitors bid (and hence p_t) is not a stationary process and more importantly is correlated with our bids. This is true even if each competitors values the ad differently and in a way specific to him. This adversarial correlation is actually one of the main differences between auctions and conventional bandit problems which has been ruled out in the paper by assuming that p_t is generated from a stationary distribution in an iid manner. 2. Assumption 1 which plays an important role in the results of the paper mentions that p_t and v_t are conditionally independent given the side information x_t. Aside from the above reason (comment 1) and even if we assume that the competitors are not learning agents, this seems a very restrictive assumption. x_t is commonly shared between all the bidders and all of them select their bids based on it. Consider for example a scenario that a group of the bidders (including us) are interested in a specific type of (target person, website) pair and they will bid higher when the associated x_t is reported. Thus, there is a correlation between those competitors bids (and hence p_t) and our bids given x_t. There is clearly a correlation between our bid and our value v_t (because we are a smart bidder) and hence, v_t and p_t cannot be independent given x_t. A few typos: 1. Assumption 1 in the Supplementary Materials is i fact Assumption 2 of the main paper. 2. There is a . missing at the end of line 313 and one at the end of Lemma 8. 3. At the end of Section C in the Supplementary Materials it should be Lemma 3 of [2] not Lemma 11 of [2].